# Using isotopes to constrain water flux and age estimates in snow-influenced catchments using the STARR (Spatially distributed Tracer-Aided Rainfall-Runoff) model

Pertti Ala-aho[1], Doerthe Tetzlaff[1], James P. McNamara[2], Hjalmar Laudon[3], Chris Soulsby[1]

[1] Northern Rivers Institute, School of Geosciences, University of Aberdeen, UK, AB24 3UF
[2] Department of Geosciences, Boise State University, Boise, ID 83725, USA
[3] Department of Forest, Ecology and Management, Swedish University of Agricultural Sciences, 90183 Umeå, Sweden.

*Correspondence to*: Pertti Ala-aho (pertti.ala-aho@abdn.ac.uk)

Tracer-aided hydrological models are increasingly used to reveal fundamentals of runoff generation processes and water travel times in catchments. Modelling studies integrating stable water isotopes as tracers are mostly based in temperate and warm climates, leaving catchments with strong snow-influences underrepresented in the literature. Such catchments are challenging, as the isotopic tracer signals in water entering the catchments as snowmelt are typically distorted from incoming precipitation

due to fractionation processes in seasonal snowpack.

We used the Spatially Distributed Tracer-Aided Rainfall-Runoff model (STARR) to simulate fluxes, storage and mixing of water and tracers, as well as estimating water ages in three long-term experimental catchments with varying degrees of snow influence and contrasting landscape characteristics. In the context of Northern catchments the sites have exceptionally long and rich datasets of hydrometric data and - most importantly - stable water isotopes for both rain and snow conditions. To

adapt the STARR model for sites with strong snow-influence, we used a novel parsimonious calculation scheme that takes into account the isotopic fractionation through snow sublimation and snow melt.

The modified STARR setup simulated the stream flows, isotope ratios and snow pack dynamics quite well in all three catchments. From this, our simulations indicated contrasting median water ages and water age distributions between catchments brought about mainly by differences in topography and soil characteristics. However, the variable degree of snow

influence in catchments also had a major influence on the stream hydrograph, storage dynamics and water age distributions, which was captured by the model. Our study suggested that snow sublimation fractionation processes can be important to include in tracer-aided modelling for catchments with seasonal snowpack, while the influence of fractionation during snowmelt could not be unequivocally shown. Our work showed the utility of isotopes to provide a proof of concept for our modelling framework in snow influenced catchments.

# 1 Introduction

Tracer-aided hydrological models provide invaluable insights into how water and solutes are partitioned, stored and transported within catchments (Seibert and McDonnell 2002, Kirchner 2006). They can also be used to explore metrics of catchment hydrological function such as water age and travel times, leading to new avenues for use in studying ecohydrological water partitioning and anthropogenic influences (Birkel and Soulsby 2015). Snow influenced catchments, particularly in the Northern hemisphere, have a long history in hydrological modelling with the ultimate goal of reproducing the stream hydrograph where spring snowmelt plays a dominant role (Hinzman and Kane 1991, Seibert 1999, Pomeroy et al. 2007). An essential part of this is correctly representing snow accumulation and melt, and importantly its spatial distribution, and as a result numerous tools of variable complexity has emerged to simulate catchment scale snow processes (Tarboton and Luce 1996, Lehning et al. 2002, Liston and Elder 2006). However, using tracer-aided models in these regions has been less common (Tetzlaff et al. 2015b) - mainly due difficulties of routine, long-term field work and sample collection in such cold and often remote regions. Nevertheless, environmental tracers have been used at some sites to increase conceptual process understanding in the field and for hydrograph separation studies (Sklash and Farvolden 1979, Laudon et al. 2002, Carey and Quinton 2004, Schmieder et al. 2016).

Stable water isotopes oxygen-18 and deuterium are commonly used environmental tracers because of their conservative properties and automatic entry to natural systems with precipitation (Birkel and Soulsby 2015). They are also increasingly easy and inexpensive to analyse in large numbers (Berman et al. 2009). In snow-influenced environments, the cryogenic processes complicate the isotope input signal through several processes. Firstly during snow accumulation snowfalls with different isotopic composition make up the snowpack with distinct isotopic layers typically persisting until the wholesale snowmelt (Rodhe 1981), though a degree of internal isotopic redistribution is typically present in the snowpack (Taylor et al. 2001, Evans et al. 2016). Internal mixing processes do not considerably influence the bulk isotopic composition of the snowpack, but fractionation due to snow sublimation has the potential to isotopically enrich the snowpack in relation to snowfall (Moser and Stichler 1974, Earman et al. 2006). Furthermore, canopy snow interception can provide an additional transient storage subjected to sublimation, and thereby fractionation processes, further amplifying the snow isotopic enrichment (Claassen and Downey 1995, Koeniger et al. 2008). Finally, several field, laboratory, and modelling studies (Shanley et al. 1995, Taylor et al. 2001, Feng et al. 2002) demonstrate how the onset of snowmelt tends to be depleted in heavy isotopes in comparison to average snowpack, and the snowpack isotopically enriches over the course of snowmelt. This "melt out" process, though a complex and variable phenomenon in field conditions, has been shown by e.g. (Taylor et al. 2002) to be rather a rule than exception for snowmelt of seasonal snow packs in various climates. As a combined results of the processes above, water entering the catchment as liquid is not only delayed in timing because of being stored as snow, but is typically also altered in its isotopic composition (Laudon et al. 2002, Schmieder et al. 2016). In most environments such processes have a high degree of spatial variability and are thus challenging to model at the catchment scale.

Historically, tracer-aided model applications have typically been lumped/semi-lumped conceptual models (Neal et al. 1988, Barnes and Bonell 1996, Hrachowitz et al. 2013, Smith et al. 2016), though some modelling studies incorporate spatial variability in the model parameterisation (Stadnyk et al. 2013, Birkel et al. 2015). Spatially and temporally limited tracer data are typically a considerable constraint in tracer-aided modelling (Delavau et al. 2017), in particular if interest lies in understanding spatially distributed flow processes over longer than event time scales. To address this challenge, the Spatially Distributed Trace-Aided Rainfall-Runoff model (STARR) was developed to fully distribute the simulation for hydrological storages, fluxes, isotope ratios and water age in the landscape (Huijgevoort et al. 2016a, Huijgevoort et al. 2016b). This follows on from previous conceptual models that have used spatially explicit frameworks for tracking isotopes in the rainfall-runoff transformation (Sayama and McDonnell 2009). The STARR model was originally developed for a long-term experimental catchment in the Scottish Highlands, the Bruntland Burn, with the aim to keep the model simple to be applicable as a generic tool across Northern regions with strong snowmelt influence (Tetzlaff et al. 2015a).

In this study, we apply the STARR model in three well-established research catchments with long and frequent datasets of stable water isotopes (Tetzlaff et al. 2017). All experimental catchments experience seasonal snow influence, but are contrasting in their topography, dominant soil types and canopy cover. The main advancement of the STARR model reported here is replacing the original degree-day snow module with an energy-driven process-based snow module that can track isotopes (Ala-aho et al. 2017b). The novel snow module encompasses original algorithms to account for (1) sublimation fractionation of snow isotopes of canopy intercepted snow and ground snow pack and (2) time-variant depletion of the snowmelt. Both processes are well documented in laboratory, field, and modelling studies (Cooper et al. 1993, Claassen and Downey 1995, Taylor et al. 2001, Laudon et al. 2002, see e.g. Carey and Quinton 2004, Koeniger et al. 2008, Schmieder et al. 2016), but not before incorporated to tracer-aided modelling in a spatially explicit manner.

The overarching goal for this study is to better understand spatial distribution and non-stationary responses of water ages in snow influenced northern catchments. We achieve this by using the STARR model to simulate spatially distributed flows, isotopes and snow water equivalent (SWE) in three long term experimental catchments. The contrasting catchment characteristics make a strong test for the model adaptability in different cold climate conditions. As a novel aspect, we test new empirical parsimonious routines for stable water isotope processes in seasonal snowpacks in our catchment scale simulations. Wider importance of our work is to advance tools for process understanding and management of snow-influenced environments that are underrepresented in research but are on the verge of drastic changes due to global climate change and economic development (Tetzlaff et al. 2015b).

## 2 Methods

### 2.1 Study sites

All three study sites are established, long-term experimental catchments with a wealth of hydrological, ecological and biogeochemical research activities associated with them. We provide a brief description of the catchments; more details are found in the cited publications.

**Fig. 1**

### 2.1.1 Krycklan C7

The most northerly (64°14' N, 10°46' E) and smallest (0.5 km$^2$) experimental catchment is Krycklan C7 (Fig. 1) in the Swedish boreal forest, approximately 50 km inland from Umeå and the Baltic sea (Laudon et al. 2013). Of the three catchments, it has the gentlest relief with altitudes ranging from 235-306 m asl. Annual average precipitation is 622 mm, approximately 35% to 50% of which falls as snow (Laudon and Löfvenius 2016). Annual average air temperature is 2.4 $^0$C with sub-zero monthly mean temperatures typically during November – March. Highest flows in the catchment are typically a result of spring snowmelt during 3-4 weeks in April/May (Fig. 2). Half of the runoff occurs during the snow-free period with higher flows towards autumn. Snow covered winters from December to March are the annual low flow season. Most of the land cover is conifer boreal forest (82%) with a mix of Scots pine (Pinus sylvestris) and Norwegian spruce (Picea abies). A small part of the catchment consist of a canopy-free minerogenic mire (18%) which is dominated by Sphagnum moss. Except for the organic soils in the mire and riparian areas adjacent to the stream, the catchment is dominated by podzolic soils. The podzolic soils are formed on compact basal till underlain by metasediments. Hydraulic conductivity of the soil decreases with depth with high transmissivity feedback initiated in the top soil with high water tables (Nyberg et al. 2001). In the forested areas with podzolic soils subsurface flow paths dominate, whereas overland flow takes place in the mire during periods of intense water inputs from rain or snowmelt (Peralta-Tapia et al. 2015a).

### 2.1.2 Bruntland Burn

The Bruntland Burn (57°8'N 3°20'E) is located in the Scottish Highlands and of the three catchments is the largest in size (3.2 km$^2$). The catchment has a wide flat glaciated valley bottom surrounded by steeper hillslopes (Fig. 1), with altitudinal relief ranging from 250 – 530 m asl. Average annual precipitation is 1000 mm, with annually reoccurring, but not dominant (typically <5% of annual precipitation), snow influence. Average air temperature is 7.0 $^0$C without any months with below negative mean temperatures. Seasonality in streamflow is much less pronounced than for the other two sites because of milder winters and lower snow influence, with peak flows typically taking place between November and February (Fig 2). Soils in the wide valley bottom riparian areas are histosols (22%) with Sphagnum spp and Molina caerulea vegetation. Organic soils in the valley bottom are underlain by glacial drift up to 30 m deep creating a considerable groundwater storage and steady

groundwater flux to riparian area (Ala-aho et al. 2017a). Steeper slopes have podzolic soils underlying by more freely draining minerogenic soils with heather (*Calluna vulgaris*) being the dominant vegetation, with patches of Scots Pine (*Pinus sylvestris*) forests. In Bruntland runoff is primarily generated within the persistently saturated riparian areas, with occasional contributions from hillslopes when they became hydrologically connected during large storm events (Tetzlaff et al. 2014).

### 2.1.3 Bogus Creek

The Bogus Creek (43°42'N 116°10'E) is the most southerly site located in the north west U.S, Idaho (Fig. 1). The catchment is similar in size to Krycklan (0.6 km$^2$), but a V-shaped fluvial valley slopes steeply from 1684 to 2135 m asl., resulting in the highest altitude catchment of the three. The site receives about 670 mm of precipitation annually, with more than 50% of it during winter as snowfall, summers being typically hot and dry. Average annual air temperature of 8.8 $^0$C is the highest among sites, but with below-zero mean monthly temperatures from November to March in the highest parts of the catchment, December to February at the catchment outlet. The stream hydrograph reflects the climate with a snowmelt influence from March to June typically peaking in May with low flows during rest of the year (Fig. 2). The soils are thin (< 1m) highly permeable sands underlain by fractured granodiorite comprising a hydrologically active bedrock groundwater storage. Bitter and choke cherry (*Prunus* spp.) and buck brush (*Ceanothus spp)* shrubs cover the most of the catchment, with a small fraction of larger trees (Douglas Fir (*Pseudotsuga menziesii*) and Ponderosa Pines (*Pinus ponderosa*)) in the valley bottom near the stream. Runoff generation at the site is subsurface-driven and runoff increases only after the water table rises sufficiently (McNamara et al. 2005).

**Fig. 2**

### 2.2 Model input and test data

The experimental catchments studied here have – in a northern context – exceptionally long and high quality data records for stable water isotopes, streamflow and meteorological variables. Isotope data from streamflow has been sampled daily for Bruntland Burn, weekly for Krycklan and more sporadically for Bogus (Fig. 3). Typical variability in stream isotopes over the year (0.05-0.95 quantiles) spans from -7.7 to -9.3 ‰ in δ$^{18}$O for Bruntland, -12.0 to -14.0 ‰ in Krycklan, and -15.6 to -17.4 ‰ in Bogus, showing seasonality with depleted values in winter months and enriched in the summer for all sites. At the Krycklan site we see a distinct snowmelt depletion during April/May. Precipitation samples were available on for most precipitation events for Bruntland and Krycklan. For Bruntland only liquid precipitation was sampled because of rarely occurring snowfall events. For Krycklan with more persistent snow cover, precipitation was sampled daily following every snow fall, melted in a cool room (+8 °C) and subsequently measured for volume using a fine graded measurement cylinder. The occasional missing values are preferentially filled by a value using a representative sample from a nearby catchment, or if not available, backfilled with data from the next available date. For Bogus, we expanded the sparse input data set applying the following methodology. We used samples from all meteorological stations in the wider Dry Creek Experimental watershed

(DCEW, n=142) to build a linear regression model to estimate continuous time series for precipitation from daily air temperature similarly as in Tappa et al. (2016). In addition, we applied an environmental lapse rate of -0.22 ‰ for $\delta^{18}$O per 100 m rise in elevation established for the DCEW (Tappa et al. 2016). Precipitation isotopes were used as model input data, isotopes in streamflow in model calibration.

**Fig. 3**

Meteorological data necessary to run the simulations are daily precipitation, air temperature, shortwave radiation, relative humidity and wind speed. For the Krycklan site, measurements from the Svartberg meteorological station 150 m southwest from the catchment outlet (elevation 10 m lower than the outlet) were used for the whole simulation period 2003-2013. For
the first two years relative humidity and wind speed data were not available, and the long term daily average was used instead. For the Bruntland Burn, averaged data from two meteorological stations installed in the catchment, one at the valley bottom (10 m above catchment outlet) and one at the top of the catchment, on the southern hillslope (210 m above the outlet) were used from July 2014; prior to that datasets from weather stations in the neighbouring Girnock catchment were utilised as in previous model application for the site (Huijgevoort et al. 2016a), with the exception of shortwave radiation which was
provided by the Centre for Environmental Data Analysis (MET Office. , 2017). For Bogus, we used the meteorological station in the DCEW, 4 km south-east and roughly the same altitude as the catchment outlet. Occasional gaps were filled with other meteorological stations in the catchment, or snow telemetry (SNOTEL) meteorological station no. 978 (National Climatic Data Center. , 2016) located 200 m north of the catchment top, 250 m above the catchment outlet. A spatially distributed environmental lapse rate of -0.6 C/100 m was applied to air temperature measurements according to the moist adiabatic lapse
rate (Goody and Yung 1995). A +5.4 %/100 m increase in precipitation was measured in the Bruntland along a hillslope covering 200 m elevation difference, and the parameter value was transferred to Bogus. We used temporally constant lapse rates, but they may vary in different seasons, latitudes, and orographic influences (Stone and Carlson 1979, Sevruk and Mieglitz 2002). Altitude effects are negligible for the gently sloping Krycklan site (Karlsen et al. 2016).

Streams for all experimental sites are gauged and quality controlled by the respective research groups and available online for
Bogus and Krycklan (Laudon et al. 2013, Boise State University. , 2017). Hourly or sub-hourly data was averaged to daily values. Snow water equivalent (SWE) in Krycklan is measured approximately 1 km west of the catchment in an open mire starting every midwinter and repeated with approximately 2-3 week intervals until the snow had melted (Laudon and Löfvenius 2016). SWE data representative of Bogus was acquired from the SNOTEL (same as for meteorological data) station where SWE is measured continuously with a pressure transducer (National Climatic Data Center, 2016). The Bruntland does not have
routine snow monitoring because of the unpredictability of the sporadic and highly transient snow influence. Streamflow and isotopes were used in the model calibration for all sites, SWE measurements for Krycklan and Bogus.

## 2.3 STARR model setup

The STARR model was first developed for the Bruntland Burn by (Huijgevoort et al. 2016a) where full details can be found. Briefly, the hydrological part of the model conceptualises soil and groundwater stores as linear reservoirs (Fig. 4) similar to the HBV model (Lindström et al. 1997). Previous, more lumped tracer-aided modelling in the Bruntland catchment (Birkel et al. 2011) was used as a basis to conceptualise the storing, mixing and routing of tracers. In addition to routing and tracking water and tracer fluxes, the mixing equations were used to estimate water ages in the stream and different conceptualised hydrological compartments of the catchment (Soulsby et al. 2015) – a feature that was also implemented in STARR. Like its predecessors, the STARR model utilises a concept of passive storage in isotopic mixing in the soil (Birkel et al. 2015). Passive storage parameterises the water stored in the soil that does not relate to changes in discharge, but increases the total mixing volume of isotopes. The major development in the STARR model presented in van Huijgevoort et al (2016a) was to spatially distribute the conceptual modelling equations to grid cells, so that each model cell has its own representation for the various storages and fluxes for both water and tracers in the conceptual model scheme (Fig. 4). Runoff fluxes from all cells are routed through the catchment to simulate the stream hydrograph and isotopic concentration. The approach allows presentation of transient water fluxes, storages and water ages in a spatially explicit way in the landscape (for a visualisation see (Huijgevoort et al. 2016b)). The STARR model is built with a modular structure in the PCRASTER PYTHON framework (Karssenberg et al. 2010). Details for the model modules and related equations and parameters are given in Appendix (A1) and (Huijgevoort et al. 2016a), with the new developments of this study described below.

**Fig. 4**

STARR was developed with an overarching goal to keep the model simple and generically applicable (Huijgevoort et al. 2016a). The major advancement to STARR presented in this paper is to make it more suitable across Northern latitudes by replacing the original degree-day snow module with an energy-driven process-based snow module with a novel capability to simulate isotopic evolution of the snowpack. The calculation routines that account for the 'water' part of the model, ground snow melt and accumulation, are based on formulations for single layer snowpack energy and mass balance equations published in (Wigmosta et al. 1994, Walter et al. 2005) and described in detail in (Ala-aho et al. 2017b). Energy balance for each time step is solved based on net radiation, latent and sensible heat, heat advection from precipitation and heat storage in the snowpack. The energy balance is coupled with mass balance equations solving the amount and ice, and liquid water retained in the snowpack and the snowmelt and sublimation fluxes. Model inputs for precipitation and air temperature are spatially distributed as described in section 2.2, and the radiation terms are adjusted to the influence of slope, aspect, hillshading, and canopy sheltering. Tree canopy snow interception and unloading are simulated after (Hedstrom and Pomeroy 1998). The isotopic ratio of the snowpack is linked to snowpack water balance simulations with the following assumptions and conceptualisations:

(1) Isotopes in the snow storage (ground snowpack and interception storage) are fully mixed within each time step. The isotopic ratio of the remaining ground snowpack is solved with the mass balance Eq. (1):

$$i_{sn_j} = \frac{i_{sn(j-1)}*SWE_{(j-1)} + i_{P_j}*S_{thru_j} + i_{P_j}*P_{liq_j} + i_{int_j}*S_{unl_j} - i_{snowE_j}*E_{snow_j} - i_{melt_j}*S_{melt_j}}{SWE_{(j-1)} + S_{thru_j} + P_{liq_j} + S_{unl_j} - E_{snow_j} - S_{melt_j}} \tag{1}$$

Where j: simulation time step; SWE [mm]: snow water equivalent in the snowpack; $i_P$ [‰]: isotope ratio in the precipitation; $S_{thru}$ [mm]: throughfall bypassing interception storage; $P_{liq}$ [mm]: liquid precipitation; $i_{int}$ [‰]: isotope ratio of snow interception storage; $S_{unl}$ [mm]: water unloaded from interception storage; $i_{snowE}$ [‰]: isotope ratio of sublimated water from eq. (1); $E_{snow}$ [mm]: amount of simulated snow sublimation; $i_{melt}$ [‰]: isotope ratio of snowmelt from eq. (2); and $S_{melt}$ [mm]: amount of snowmelt.

(2) Snow sublimation fractionates the water on the ground and the intercepted snow storage, leaving the remaining snow enriched in heavy isotopes. This is achieved by introducing an offset parameter $E_{frac}$ to determine the level of depletion in the sublimated water relative to the snowpack Eq. (2).

$$i_{snowE} = i_{sn} - E_{frac} \tag{2}$$

Where $i_{snowE}$ [‰]: isotope ratio of the sublimated water; $i_{sn}$ [‰]: isotopic concentration of the snowpack and $E_{frac}$ [‰]: offset parameter. $E_{frac}$ is calibrated and allowed to take values between 0 … 15 ‰ based on the equilibrium difference of 15 ‰ between ice and vapour isotopic ratio at 0 $^0$C temperatures (Ellehoj et al. 2013).

(3) Water leaving the snowpack is initially depleted in heavy isotopes with respect to the snowpack, and the snowmelt water grows progressively more enriched as the snowmelt advances by with the Eq. (3). The empirical formulation in Eq. (3) is proposed in order to mimic the gradual isotopic enrichment of both snowmelt runoff and snowpack over the overall melt period, which is frequently observed in field studies (Taylor et al. 2002) and theoretically show in modelling experiments (Feng et al. 2002)

$$i_{melt} = i_{sn} - \frac{M_{frac}}{d_{melt}} \tag{3}$$

Where $i_{melt}$ [‰] is the isotope ratio of the snowmelt water, $M_{frac}$ [‰] is the offset parameter and $d_{melt}$ is the number of days snowpack has experienced snowmelt. $M_{frac}$ is calibrated and allowed to take values between 0 … 3.5 ‰ based on the equilibrium difference of 3.5 ‰ between ice and liquid water isotopic ratio. Using the equation the first melt event is offset by the value of parameter $M_{frac}$ (the value for $d_{melt}$ being 1), and with subsequent snowmelt days the offset approaches 0 with increasingly higher values for $d_{melt}$.

For full details how SWE, $S_{thru}$, $S_{uln}$, $E_{snow}$ and $S_{melt}$ are simulated, the reader is referred to Ala-aho et al. (2017b) where the snow routine algorithms are provided with a modelling experiment to demonstrate the snow algorithm functionalities and a field test of the model performance is done against snowmelt lysimeter data. The new process-based energy balance snow module is embedded in STARR similarly to the previous degree-day module, providing water and isotope influx to soil storage. Snow age is tracked according to Eq. (1) but replacing the isotope ratios with water ages in all storages and fluxes. Water age of incoming precipitation is taken as 1, and the sublimation and melt fractionation processes do not alter snow age. With the

full-mixing assumption, water stored as snow is aged while the snowpack persists, but this is refreshed with new snowfall, weighted by snow amount. Water entering the catchment during snowmelt will therefore be reasonably approximated as having an age younger than the full snow-covered season, but considerably older than only the most recent snowfall. Therefore the snowmelt entering the catchment is typically older than precipitation, depending on the length of season of snow-coverage.

A second modification was to reformulate the soil storage parameterisation and to change the storage-discharge conceptualisation from a linear reservoir into a power law (Eq. A12). The concept of field capacity also is changed from Huijgevoort et al (2017a) from where the field capacity (FC) was defined as the maximum amount of water that could be stored in the linear soil storage (SM), whereas now we conceptualised this – as more typically done in soil physics - as the amount of water that is preferably retained in the soil, defined by parameters for volumetric field capacity and soil depth (Eq.
A9), both technically measurable in the field. . These changes were done to create a more physically based parameterisation for the model and to allow nonlinear seepage and outflow processes with high soil storage values. The need for adaptation became apparent when applying the original model algorithms to Krycklan and Bogus. In its original formulation the model did not allow for high enough seepage rates from the soil to groundwater domain as observed in Bogus, or non-linearly increased runoff generation from the soil domain during times of high soil storage, also known as the transmissivity feedback,
present in Krycklan.

Finally, in contrast to previous work, here we assumed the evaporation age to be equal to the water age in the soil storage. In the previous model implementation in Huijgevoort et al. (2016a), the simulated soil water age was affected by evaporation, but the simulated isotope composition of the soil was not; as a result the simulated evaporation age was not informed/constrained by the isotope model calibration and in this study it was excluded to simplify the model. Re-incorporating
evaporated water age in the simulations would benefit from vertically layered soil parameterisation and from explicit hydrological partitioning between evaporation and transpiration (see e.g. Sprenger et al. 2016a).The mathematical formulations of the core functions of the STARR model with the soil model alterations explained above are given in Appendix (A1) and (Huijgevoort et al. 2016a).

## 2.4 Model parameterisation and calibration

We used the Monte Carlo approach in model calibration and carried out 10 000 simulations for each experimental site to test the model performance with varied parameter sets. We used random sampling of the parameter space assigning uniform distribution for all parameters, with pre-defined parameter ranges (minimum and maximum parameter value) given in Table 1. The model was run 10 000 times, each run with a different, randomly sampled parameter set. Because some of the parameters were not measurable in the field or could not be estimated from the literature, we did a series of preliminary simulations for
all sites to find appropriate sampling ranges for these parameters.

For the snow module we included four parameters in the Monte Carlo calibration (Table 1): Correction coefficient for under catching snowfall ($c_{corr}$), threshold temperatures for precipitation phase ($TT_{low}$ below which all rain is considered as snow and $TT_{high}$ above which all rain is considered as liquid), and an empirical coefficient for decreasing snow albedo for ageing

snowpacks ($a_{pow}$). For details about the calibrated parameter see (A1) and the equations for the energy-balance based snow module see Ala-aho et al. (2017b).

For the soil module we included parameters influencing the amount of potential storage in the soil ($S_d$), and parameters ($k_s$ and $k_{pow}$) controlling the amount of outflow from the soil storage ($Q_{soil}$). For the Bogus site, seepage from soil to groundwater storage proved an important process. Therefore, parameters controlling seepage ($\beta_{seep}$ and $\beta_{pow}$) were included as calibrated parameters. For field capacity and porosity we used site-specific parameters values based on prior research in the catchments (Nyberg et al. 2001, McNamara et al. 2005, Ala-aho et al. 2017a). For Bogus, we used a soil depth map derived for the whole DCEW (Tesfa et al., 2009) for spatially distributed soil depth values. The organic soils for the riparian peatlands in Bruntland and Krycklan were parameterised according to (Päivänen 1973) in terms of porosity and field capacity (FC). For the groundwater one parameter ($k_g$), which linearly relates the groundwater storage to groundwater outflow ($Q_{gw}$), was calibrated. Finally, three parameters affecting the routing of isotopes were included in the calibration: $E_{frac}$ setting the offset between sublimated and remaining isotopic ratio in the snowpack (see Eq. 2), $M_{frac}$ regulating the isotopic fractionation between snowmelt water and snowpack and $SM_{pas}$ representing an additional mixing volume in the soil storage required to dampen the isotopic variability of the soil water.

Initial conditions for the Monte Carlo model runs at each site were established by looping the input data until the storages and water ages appeared to plateau at a constant level. State variables (water storages, isotope ratios, water ages) in the groundwater module were the only variables with high storage values which were not reset during the annual water cycle, and therefore most influenced by the looping. The spatially distributed values for groundwater storage, isotope ratios and water ages received after the looping were used as initial conditions in the subsequent Monte Carlo runs. For each individual Monte Carlo model run, a spin-up period of one year was used for Bruntland and Krycklan, and of two years for Bogus, to reduce the impact of initial conditions on the calibration period. For Bruntland, we looped climate data for the first year of the simulation for spin-up input. For Krycklan and Bogus, we used the climate data from previous years. A two year period was used in Bogus because the initial storages were greater and therefore stabilising the storages at the beginning of each Monte Carlo run was considered to take a longer time.

During the model optimisation we wanted to keep the calibration setup and objective functions identical across catchments to make the results comparable. We used a single goodness-of-fit (GOF) metric for each observation-simulation pair (streamflow, isotope ratios, SWE) to differentiate between rejected model runs and those accepted as "behavioural". The Kling-Gupta efficiency (KGE, Eq. (4)) was selected as GOF for streamflow and SWE and mean absolute error (MAE. Eq. 5) for the isotope ratios.

$$KGE = 1 - \sqrt{(r-1)^2 + (\mu_s/\mu_o - 1)^2 + (\sigma_s/\sigma_o - 1)^2} \tag{4}$$

$$MAE = \frac{\sum_{i=1}^{N}|s_i - o_i|}{N} \tag{5}$$

Where r: Pearson correlation coefficient; μ: the mean; σ: the standard deviation; subsripts s and o refer to simulated and observed values, respectively, N: number of simulation-observation pairs

Kling-Gupta efficiency was used because it combines several measures of misfit between observations and simulations (correlation, bias and a measure of relative variability; first, second and third term inside the square root in Eq. 4, respectively) into a single number in a more robust way than the frequently used Nash-Sutcliffe performance metric (Gupta et al. 2009). Additionally, the NSE puts a primacy on simulation of high flows, whereas, for a hydrological model to accurately and simultaneously capture isotope dynamics across the flow regime, a more balanced GOF measure for stream flows is needed as shown in other studies (e.g. Birkel et al., 2015)". Mean absolute error was used for the isotopes because it focuses on minimising the bias between observations and simulations and the relatively low number of stream isotope samples in Bogus did not allow to use a measure placing emphasis on variability and correlation. In addition, MAE gives an intuitive number of typical prediction error easily comparable to, for example, the analytical error of the isotope ratio. The selected GOF metric ultimately remains a subjective choice in any model calibration, but with the considerations above we found the selected metrics facilitating convenient and robust comparison between catchments.

From the ensemble of 10 000 model runs, the 100 "best" runs were identified using the cumulative distribution function (CDF) of the GOF measures with the following logic: Each parameter set, and the resulting simulation output, maps a value on the CDF of the GOF measure for all three calibration variables (KGE for streamflow and SWE; MAE for isotope ratios). We determined a threshold quantile in the GOF CDF's above which the GOF from exactly 100 runs in all calibration targets were mapped using:

$$F_{Xf}(x_f) = P\big(GOF_f \leq x_f \mid n(GOF_f \geq x_f) = n_{run}\big) \tag{6}$$

$$F_{Xi}(x_i) = 1 - P\big(GOF_i \leq (x_i) \mid n(GOF_i \leq x_i) = n_{run}\big) \tag{7}$$

$$F_{Xs}(x_s) = P\big(GOF_s \leq x_s \mid n(GOF_s \geq x_s) = n_{run}\big) \tag{8}$$

Where subscripts f, i and s represent flow, isotope and snow, respectively; $F_X(x)$ is the threshold quantile; GOF is the goodness of fit measure; n() is the number of GOF samples located above (or below in Eq. 7) the GOF value x in the CDF; $n_{run}$ is the specified number of runs (in our case 100). Values for $x_f$, $x_i$ and $x_s$ were obtained using an iterative algorithm that satisfies $F_{Xf}(x_f) = F_{Xi}(x_i) = F_{Xs}(x_s)$ when $n_{run}=100$, in a way that maximises the $F_X(x)$.

To clarify the calibration procedure with an example, let's consider two GOF measures, KGE of streamflow and SWE, to constrain the selection of 100 behavioural simulations from an ensemble. In this case it is unlikely, although possible, that the same 100 simulations that produce the highest GOF values for streamflow would also have the highest GOF values for SWE. To find the threshold quantile above which the GOF from exactly 100 runs in both calibration objectives map, first an initial guess is made; we used $F_{Xf}(x_f) = F_{Xs}(x_s) = 0.5$, which corresponds to the median of GOF values for $x_f$ and $x_s$, streamflow and SWE, respectively. This quantile as a threshold it is checked how many individual simulations produce GOF values that are higher than $x_f$ and $x_s$ for *both* streamflow *and* SWE, respectively. If the number of simulations above the $x_f$ and $x_s$ GOF thresholds in both objectives is higher than the preassigned number $n_{run}$ (in our case 100), a step up the CDF is taken, by adding a small increment in the threshold quantile, for example: $F_{Xf}(x_f) = F_{Xs}(x_s) = 0.51$. Then the number of simulations for which

KGE value is exceeded $x_f$ and $x_s$ for *both* streamflow *and* SWE are again counted for the updated threshold, and the process is repeated, until a quantile $F_{Xf}(x_f) = F_{Xs}(x_s)$, for which $n_{run}=100$ is reached. The resulting threshold GOF value $x_f$, in this example measured in KGE, will be lower than if constrained by flow data alone, because some simulations producing a good KGE for flows will be rejected as they have a $GOF_f < x_s$ for SWE.

Because for the KGE high numbers indicate a good model fit whereas for MAE low numbers indicate a good fit, for the isotope fit the threshold MAE quantile was calculated as $1 - F_X(x)$. For Bruntland, only equations 6 and 7 were used because SWE data was not available and snowmelt rarely dominates the hydrograph. The simulations for which GOF measures were above (below for MAE) the threshold in all calibration objectives were retained as the 100 best, behavioural runs. Reported ranges for the GOF measured are obtained using: $[F_X^{-1}(min(x_{ret})) , F_X^{-1}(max(x_{ret}))]$ where $x_{ret}$ is the array of quantiles of the retained

runs. The introduced approach allows pre-specifying the number of behavioural runs while circumventing the need to combine the GOF metrics into a single objective function (e.g. (Huijgevoort et al. 2016a). When a single objective function is constructed from multiple GOF metrics, it is often difficult to combine GOF metrics that need to be maximised (such as KGE) and minimised (such as MAE) in the model calibration. Our approach is based on quantiles of the GOF metric rather than its numerical value, making the method convenient in combining metrics that are to be minimised or maximised, or have different

ranges of numerical values.

We explored the sensitivity of parameters involved in the behavioural runs by calculating the ratio between pre and post calibration standard deviation of the parameter values. If the parameters for behavioural runs occupy a smaller range than the sampled parameter values, (range approximated as standard deviation), this will result in a ratio of less than one. We used this ratio as a simple proxy for parameter sensitivities in order to identify the most sensitive parameters and to compare parameter

sensitivities across catchments. A similar sensitivity analysis for each site was done prior to the final calibration runs in order to exclude insensitive parameters from the calibration and to look for suitable ranges for the sensitive ones.

## 3 Results

### 3.1 Simulation of time series of streamflow, stream isotope ratios, SWE and water ages

The model is able to produce a good fit to all calibration objectives - that is streamflow, stream isotope ratios and SWE - for

all three catchments (Figs. 5, 6 and 7). For the three sites, the GOF metrics for streamflow KGE range between 0.5-0.8 and mean absolute error in stream isotopes ranges between 0.3 – 0.5 ‰.

For Krycklan (Fig 5.), fits for the stream flows span KGE values from 0.49 to 0.82. In general, low flows are matched very well, as well as the timing and magnitude of snowmelt-induced runoff. The only notable bias is that the flow peaks during summers are not always fully captured. The model reproduces both the stream isotopic depletion due to snow melt and summer

enrichment caused by more enriched rainfall with good accuracy, and the absolute error (0.35 – 0.5 ‰) being close to the analytical accuracy of 0.2 ‰. The most consistent bias is seen during winter, when the simulations are not able to capture the gradual depletion of stream isotopes. Snow amounts are simulated reasonably well with KGE's up to 0.72, with the exception

of winter 2007. Failure to simulate the amount of accumulated snow has immediate consequences producing poor fits also for streamflow and stream isotopes. Simulated water ages show seasonality with progressively aging water during the winter (~2-3 years), a decrease in age (~ 1-2 months) during snowmelt, and greater variability during the summer with water sources being a mix of older water from groundwater storage and younger water entering and exiting the soil storage during and after storm events.

**Fig. 5**

Streamflow for Bruntland Burn (Fig. 6) is generally well captured for both dry and wet seasons with consistently high KGE's between 0.74 – 0.79. A similar problem as in Krycklan seems to occur and the magnitudes of some of the largest events are underestimated. Simulations of $\delta^{18}O$ capture the more gradual trends of depleted winters and enriched summers, with a consistently low absolute error (0.31 – 0.41 ‰), though some more marked isotope excursions in larger events (e.g. winter of 2015/2016) are under-estimated. Snow influence, when plotted for comparative purposes in the same scale as the other experimental catchments, is minor, but still present in some winters and the flows of melt events (e.g. in winter of 2012/13) are captured. Stream water ages in Bruntland are more strongly influenced by seasonality in precipitation with virtually no snowmelt influence. Oldest ages (~4-5 years) occur during dry summer periods, particularly prominent in 2013 and 2014. Wetter winter periods seasons result in windows of younger water dominance (~2-6 months) at high flows, as seen in 2013/2014 and 2015/2016. However, large summer events can have similar younger waters.

**Fig. 6**

The hydrological regime of Bogus (Fig. 7) is heavily dominated by the spring and occasional mid-winter snowmelt, with very little variability in flow during dry and hot summers. These dynamics are markedly different to the other two catchments, and are adequately reproduced by the model with KGE's between 0.56 – 0.78. The low variability in the stream isotope response is for most parts enveloped by the model outputs, with a slightly higher range for absolute error than for the two other sites (0.27-0.64 ‰). Due to the lower number of streamflow and precipitation isotope samples, the tracer time series has less power in constraining the model, reflected in higher uncertainties around isotopes and water ages. Dynamics of snow accumulation and melt are well captured (KGE's between 0.76 – 0.88), though there is a tendency to underestimate the snow amount. Simulated water ages are considerably older than for the other two catchments. The age is reduced annually most significantly by the snowmelt but only down to 0.5 - 1 years (while in Krycklan and Bruntland the ages are in ranges of weeks and months, respectively), followed by a baseflow period in the summer with older water (>10 years) simulated from the groundwater storage. Autumn rainfalls again reduces the water age, from where the ages are occasionally refreshed by mid-winter snowmelt events, as for example in winter of 2010/2011.

**Fig. 7**

### 3.2 Simulated water ages – comparison of spatial distribution and PDF's between catchments

The PDFs of the simulated water ages (best 100 runs, Fig. 8) for Krycklan and Bruntland show a similar age distribution skewed towards younger waters. Water ages in Krycklan are younger than in Bruntland with a modal water age ~ 3 months

and younger median age (< 1 year). In Bruntland, the stream water age distribution shows less kurtosis with a mode of ~14 months and median age of ~ 1.5 years. The water age distribution for Bogus is very different from the other two with a flatter and wider distribution and older median age of ~ 5 years with a largely missing young water (< 0.5 year) component.

**Fig. 8**

The spatial distribution of the water ages for dry and wet conditions are demonstrated in Fig. 9 and through an animation (S1). First looking into the water ages across catchments in dry conditions (top row in Fig. 9), water in Krycklan is typically younger than 1.5 years, with older water (>2 years) primarily focused in near topographical lows where older water accumulates because of converging groundwater flow. In Bruntland there is a separation between young water (<1.5 years) on the hillslopes and old water (>4 years) in the valley bottom areas. In Bogus the water age is predominantly >4 years, except for some

individual cells with low storage on local topographic highs. To contrast spatially distribution of water age during wet conditions (bottom row in Fig. 9), Krycklan experiences the smallest change visually, mainly because the water throughout the catchment was relatively young to start with. In Bruntland there is a substantial decrease in age for both hillslopes (down to <0.5 years) and valley bottom (<1.5 years). In Bogus water age is brought down in areas with young water input from snowmelt - in the particular snapshot above elevation 120 m from the outlet (compare with Fig. 1).

**Fig. 9**

The intimate relationship between spatially distributed runoff and water age is best seen in the animation provided in the supplementary material (S1). The animation shows for each model cell its water age (top row) and runoff (middle row), and the simulated stream hydrograph at the outlet (bottom row). Simulations span one calendar year and a typical hydrological year is shown for each catchment (2008-2009 for Krycklan, 2014-2015 for Bruntland and 2011-2012 for Bogus). A threshold

of 0.1 mm for runoff is selected to highlight the spatial variability in runoff generation. The temporal variability in both water age and runoff demonstrates the dominant snow influence in Krycklan and Bogus in terms of producing larger fluxes of young water to the streams. The main difference between the two catchments is that in Krycklan snowmelt takes place at the same time in the entire catchment over a period of three weeks, while in Bogus the snowmelt proceeds from lower to higher elevations over a period of almost two months. Both catchments show only minor variability in water age and runoff outside

the snowmelt period. In Bruntland, the spatially distributed response is driven by rainfall events throughout the year. This is seen by constantly expanding and contracting areas of runoff generation and a refreshing of the water age in the valley bottom due to rainfall events.

The simulated storages in soil and groundwater (Table 2) provide an explanation for the differences between simulated water ages in the catchments. The median values for storages are almost an order of magnitude higher from Krycklan (64 mm) to

Bruntland (540 mm) to Bogus (2182 mm), with median ages increasing in the same order. Differences in soil storage are much lower and for Krycklan and Bruntland in particular help explain the dominance of younger waters at high flows due to the limited mixing potential in groundwater storage. In Krycklan passive soil storage dominates the isotope mixing over water storage soil and groundwater. In Bruntland the mixing volumes in passive storage and soil and groundwater are approximately

equal in magnitude, whereas in Bogus mixing volume available in the groundwater store greatly exceeds the soil and passive storage.

## 3.3 Parameter sensitivities

Based on the sensitivity analysis (Fig. 10) a number of the model parameters can be interpreted as insensitive for all sites, namely ones controlling snowfall threshold temperatures ($TT_{low}$ and $TT_{high}$) snowmelt fractionation ($M_{frac}$) and soil depth ($S_d$). For other parameters, the sensitivities vary between sites depending on the dominant runoff generation processes. For the catchments with a strong snow influence and SWE data included in the calibration, i.e. Krycklan and Bogus, parameters related to snow sublimation fractionation ($E_{frac}$), correction coefficient for snowfall ($c_{corr}$) and the parameter for an ageing snow albedo ($a_{pow}$) demonstrate sensitivity. Passive storage for isotope mixing volumes in the soil ($SM_{pas}$) is sensitive for Bruntland and Krycklan where outflow from soil storage plays an important role in isotope simulation, but not that important for the GW dominated Bogus.

**Fig. 10**

For parameters controlling the outflow from soil storage, $ks$ and $ksPow$, there appears to be a trade-off in sensitivity for all sites: if the one is sensitive, the other shows less sensitivity. The parameter relating groundwater flux to storage $k_g$ is sensitive for the sites with larger GW storage (Bruntland and Bogus), but not so much in Krycklan with less groundwater storage (see also Table 2).

## 4 Discussion

### 4.1 Simulations of streamflow, stream isotopes and snow in northern headwaters

This study demonstrates the flexibility and generality of the STARR modelling framework now advanced for snow melt isotope routines to facilitate simulations in northern snow-influenced catchments. Correctly representing snow and runoff generation processes in northern catchments is important in hydrological modelling and will be increasingly important with a warming climate and following changes in cryogenic processes (Barnett et al. 2005, Berghuijs et al. 2014). Our results show reasonable performance in terms of GOF metrics. The streamflow KGE was around 0.6-0.8, KGE for SWE between generally 0.6-0.8 and mean absolute error in isotopes around 0.3 – 0.5 ‰. The good fits in all three calibration objectives (streamflow, stream isotopes tracers and SWE) are particularly noteworthy because we used exceptionally high quality time series (5 to 10 years) – in the context of northern, snow-influenced headwaters - from three long term experimental watersheds, which inevitably encompass a range of different hydrological conditions and extremes that the model needs to capture. The fact that the model can simultaneously satisfy three calibration objectives over long time periods gives confidence in the model realisations and the resulting water age estimates (McDonnell and Beven 2014). The spatially distributed model structure would allow further model testing using internal model variables, such as soil and groundwater and snowmelt isotope

composition, as done successfully for the sites in Huijgevoort at al. (2016a) and Ala-aho et al. (2017b), respectively. However, with the focus on catchment comparison in this study we restrict our analysis to the stream isotopes, which have been sampled in all study sites.

The Kryckan catchment was a strong test for the model adaptability to snow-influenced environments because of its reoccurring winter and exceptionally high quality long term isotope datasets for snow precipitation, streamflow and SWE. Moreover, the switch to summer rainfall-dominated events provided an additional challenge. The model, in general, performed well, but the misfit in winter 2007 gives an interesting insight into model failure (Andréassian et al. 2010). That year was exceptional in the data record exhibiting an intensive snowmelt event in late 2006 (Fig. 5) and almost 50% of snow was lost through snow interception sublimation over the course of winter (Kozii et al. in review). This was not fully captured by the model and led to overestimation of the simulated snow amount and delayed timing of snowmelt, which in turn resulted in overly depleted stream isotope ratios and overestimated streamflow (Fig. 5). The model's injection of an over-estimated flux of depleted snowmelt water into the catchment likely has a carry-over effect underestimating streamflow isotopes through the year 2007 and early 2008. This exceptional event illustrates how intimately snow processes are linked not only to simulating water, but also, crucially, isotope fluxes in northern catchments (Stadnyk et al. 2013, Smith et al. 2016).

For the Bruntland Burn, similar model fits to van Huijgevoort et al. (2016a) were achieved, despite some modifications in the code. This was expected because using the same input data for the first three out of five simulation years, but the additional years encompassed extreme events such as a 200 year return period flood at the turn of 2015-2016 giving the model a substantial challenge. Notable misfits in isotope ratios are present in the summer periods of 2014 and 2015 which may be caused by evaporative fractionation from pools in the riparian peat areas – a process not currently included in the model algorithms but one that has been shown important in the catchment (Sprenger et al. 2016b). Smith et al. (2016) have successfully included the soil evaporative fractionation in their spatially distributed tracer-aided simulations, and similar approaches could be adopted to the STARR model to improve model realism during summer periods with elevated evaporation. For the Bogus catchment, less frequent isotope data for precipitation and streamflow were available, but we argue that the available data, though fewer than for other sites, was still very useful in model calibration. In Bogus, both calibration data for stream flow and isotope forced the model to route much of the water through deep flow paths, mobilizing the groundwater storage. Streamflow data enforced a prolonged and damped response to snowmelt. To reproduce this, the model has to divert high volumes of snowmelt water entering the catchment in a short time window without an excessive stream response, requiring high seepage rates to groundwater storage. In addition, measured stream isotope ratios reside close to average winter precipitation (-16 … -17 ‰), inferring a lot of snowmelt recharge and mixing in the subsurface. The model behaviour fits well with the conceptual understanding of the streamflow generation processes in the catchment, where runoff takes place only after the water storage in the subsurface accumulates sufficiently (McNamara et al. 2005, Kelleners et al. 2010). Even though snow cover at the top of the catchment is persistent throughout winter, the catchment lies in the rain-snow transition zone and lower parts of the catchment are likely to experience midwinter snowmelt and rain on snow events (Kormos et al. 2014, Evans et al. 2016). The spatially distributed process-based snow routine developed in Ala-aho et al. (2017b) is able to represent the

spatial differences in snow accumulation and melt seen in reproducing the midwinter flow peaks in winters 2008/2009 and 2010/2011.

We used 11 parameters in the model calibration at all sites, some of which were distributed for different soil units in Bruntland and Krycklan to reflect differences in soil characteristics and readily established conceptual models for the catchments' runoff generation (Nyberg et al. 2001, Laudon et al. 2004, Birkel et al. 2014, Tetzlaff et al. 2014). Based on our sensitivity analysis we already excluded a number of insensitive model parameters from the calibration process. The parameter sensitivity is not fully comparable across sites, because we decided on slightly different calibration ranges for parameters *sdepth*, *ks* and *ksPow* based on tentative model testing to account for the substantial differences in catchment characteristics and resulting dominant runoff generation processes. We wanted to focus on sampling the "behavioural" ranges of the parameters to improve our chances for good model fits.

## 4.2 Spatially distributed water ages reveal runoff generation mechanisms

The methodology in STARR to track spatially distributed water ages proved very insightful in highlighting differences between catchments and their flow processes. The model allows for water source appointment, i.e. in what part of the catchment and from which model compartment the runoff is being generated (animation S1), comparable to information that can be gained from data driven hydrograph separation techniques (Rodhe 1981, Laudon et al. 2002, He et al. 2015). Estimates of stream water age give comparative, integrated signals of what runoff generation processes dominate in the catchment and the timing of their seasonally activation. Such spatially distributed estimates of water ages and their temporal evolution are still limited (Sayama and McDonnell 2009), in particular in data sparse northern regions. Other techniques for water age estimations such as transit time distributions (McGuire and McDonnell 2006), lumped conceptual modelling (Soulsby et al. 2015) or more recent storage selection functions (Rinaldo et al. 2015) do not allow such spatially detailed insight into catchment functioning. Water age distributions highlight the differences between catchments (Fig. 8) whereas the age time series (Figs. 5, 6, 7) give insights into runoff generation processes within individual catchments. Spatially distributed animations of the model outputs (S1) provide an intuitive visual tool for both comparison across catchments and identifying spatio-temporal dynamics of runoff generation processes within individual catchments.

The most striking feature when comparing water age probability densities (Fig. 8) is the difference between Bogus and the other sites: both the flat shape of the distribution and old median water age are reflective of long residence time in Bogus. Water age distributions for Bruntland and Krycklan have a heavy-tailed shape commonly encountered in catchment water age or transit time distributions (McGuire and McDonnell 2006). The main difference between the two is the higher fraction of young (<1 year) water in Krycklan which can be explained by the lower subsurface storage (Table 2), a strong influence of relatively young snowmelt water (S1), and smaller catchment area (Fig. 1). However, the youngest waters in Krycklan are not from snowmelt events, because snowpack is aging throughout the winter, thus, giving the snowmelt water a typical age of 2-3 months, coinciding with the mode age in Krycklan (Fig 8). Median water age for Krycklan is ~ 11 months, which is slightly below an independent estimate of 1.5 – 2 years for catchment mean transit time using the convolution integral method to fit a

gamma distribution to same isotope dataset (Peralta-Tapia et al. 2016). In Bruntland, the median age of stream water was ~ 1.5 years, which is close to the previous STARR application to the site~ 1.6 years in Huijgevoort et al. (2016a) and in broad agreement with two independent modelling studies yielding median water ages of ~ 0.9 and ~ 1.8 years (Soulsby et al. 2015, Benettin et al. in press), respectively, and convolution integral based mean transit time estimate of 1.9 years (Hrachowitz et al. 2010).

Time series for water ages (Figs. 5, 6, 7) and the spatially distributed animation for both ages and runoff (Fig. 9 and S1) reveal snow-dominated flow regimes for Krycklan and Bogus with a distinct winter baseflow period with old water, followed by a period of stream water rejuvenation during snowmelt. The oldest water ages in both catchments occur during summer, not winter base flow. We attribute this to active evaporation processes in the model drawing water from groundwater storage (with old age) via capillary flow. Water from GW storage mixes with the water stored in soil storage, aging the storage. Some of this soil water is routed to the stream (Fig. 4), resulting in old stream water ages. In Bruntland, the water ages are much more event driven reflecting less marked seasonality that is not dictated by snow but regular transitions between drier and wetter periods (S1).

We used spatially varied parameterisation for soil properties and vegetation where there was sufficient data to do so; a differentiation between mineral and organic soil was made in Bruntland and Krycklan, a detailed soil depth map was used in Bogus and vegetation LAI was estimated form either vegetation maps (Krycklan) or three height (Bruntland and Bogus). Even so, naturally occurring small-scale heterogeneity is known to influence the catchment hydrological response (Beven and Germann 1982), but it is difficult to represent in hydrological models - one of the persistant problems in hydrological modelling (Blöschl and Sivapalan 1995, Beven 2002). Every new introduced element of heterogeneity typically comes with a burden of increased number of parameters (see soil parameterisation in Table 1) which can lead to model equifinality issues (Beven 2006). We opted to minimise the number of calibrated parameters, with the trade-off off bringing spatial variability in parameter values only when supported by field data.

Another parameterisation issue in our work rises from specifying initial conditions for the groundwater storage for the Monte-Carlo runs. If the initial GW storage is not in "balance" with the magnitude of the outflow coefficient ($k_g$), which is randomly varied in the calibration, it can lead to GW storage reduction or increase over time. Our simulations at the Krycklan site show symptoms of such imbalances between the $k_g$ parameter and the initial GW storage, as the variability and median in the simulated stream water age declines over the ten year period (Fig. 5). The non-stationarity in age suggest that the groundwater influence (GW storage has older water) reduces over time. In further analysis (data not show) in most of the behavioural simulations the total GW storage in Krycklan in fact grows smaller over time. A longer spin-up period for the Krycklan simulations would alleviate the issue, with the burden of increased runtimes. In addition, even though our simulations for streamflow during winter is well captured (Fig. 5), the isotope composition in some winters does not shift adequately towards more depleted values (isotopes in deep groundwater between -13 and -14 ‰ (Peralta-Tapia et al. 2015b)), suggesting a too low groundwater contribution. The misfit in winter isotopes suggests that the model has problems in switching from soil source to a more depleted groundwater source during winter. It should be pointed out, that such analysis and insights are only possible

because of the ability of the STARR model to simulate stable water isotopes and water ages – these issues would not become apparent if using only streamflow hydrograph to evaluate the model performance.

## 4.3 Algorithms for isotope fractionation snow are crucial for simulating isotope dynamics in northern regions

To our knowledge, isotopic fractionation from snowpack sublimation (intercepted and ground snow) and snow melt has not been considered in previous tracer-aided models, though it has been identified as a potentially important process to improve model realism for snow influenced environments (Fekete et al. 2006, Smith et al. 2016, Delavau et al. 2017). In this work, the novel algorithms for spatially distributed snow isotope routines incorporating the processes above, were most extensively tested in the Krycklan simulations. Annually reoccurring snowmelt depletion of streamflow was generally well captured by the model (except for 2007 as previously discussed), often even for the subtle nuances in the depth of depletion peak between years (Fig. 5).

Isotopic fractionation of the snowpack caused by sublimation (see Eq. 2) was essential to include to capture the isotopic enrichment of snowpack during the winter for Krycklan. This is to some extent evident in the high sensitivity of the $E_{frac}$ parameter in Fig. 10, though sensitivity as approximated here does not reveal what parameter values produced good model fits. For Krycklan, the range for the behaviour simulations was 5-15 ‰ with a median of 8.6 ‰, which means an sublimative offset was beneficial in enriching the snowpack in heavy isotopes and eventually matching the observed streamflow isotope levels, agreeing with findings in Laudon et al. (2004) and Laudon et al. (2007). On the contrary, though the $E_{frac}$ parameter was also sensitive in Bogus, it displayed a typical range of 0-5 ‰ with a median of 2.2 ‰, suggesting a need for the model calibration to minimise the impact of snow sublimation fractionation to gain good model fits in streamflow. This agrees with the findings of (Evans et al. 2016) who did not find evidence of sublimation fractionation in their detailed measurements of snowpack evolution over one winter in the same catchment. It should be noted that the input data for snowfall for Bogus site was not as comprehensive as for Krycklan, therefore, the additional uncertainty in the model input may mask the sublimation fractionation effects, if present. However, our snowfall inputs were similar to ones estimated for the area in (Tappa et al. 2016) and therefore, on average, likely not too far off. The $E_{frac}$ parameter was insensitive for the Bruntland (Fig. 10), which is not surprising given the considerably smaller snow-influence compared to the other two sites (Fig. 6).

In our parsimonious snow isotope simulation approach we did not differentiate between kinetic and equilibrium fractionation in snow sublimation, and we only simulated only the $\delta^{18}O$ isotope because of better data availability in all sites. This simplification prevented us from simulating additional isotopic indices for evaporation, such as the d-excess (Dansgaard 1964), that would indicate deviations from the meteoric water caused by kinetic fractionation. In typical winter conditions with low air temperature and high relative humidity, we would expect the equilibrium fractionation to dominate over kinetic fractionation (Gat and Gonfiantini 1981), therefore making weather conditions and the differentiation between the two processes of lesser importance. We also did not differentiate between sublimation (ice to vapour) and evaporation (liquid water retained in the snow to vapour). Liquid water evaporation has a smaller equilibrium fractionation factor (3.5 ‰) compared to sublimation (15 ‰), so separating the different processes could lead to smaller simulated fractionation signal. In our approach

we lumped the above fractionation processes and their temporal variability caused by meteorological conditions in the $E_{frac}$ calibration parameter with the purpose of keeping the simulated isotope process complexity to a minimum, which is in line with our conceptual modelling of water in the catchment. The simplified approach is further justified by the limited power of the validation data (isotopes in streamflow) to constrain the additional parameters required for more sophisticated snowpack isotope modelling methods (Taylor et al. 2001).

The parameter $M_{frac}$ that adjusts the depletion of initial snowmelt water and progressive enrichment through the snowmelt (see Eq. 3) did not appear to exhibit sensitivity in relation to model outputs (Fig. 10). In the model setups there was a strong component of water isotope mixing the soil and/or groundwater storage, with high values of passive storage (behavioural parameter ranges for $SM_{pas}$ between 250-300 in Krycklan and Bruntland, Table 2) which likely masked the detailed dynamics of a spatially and temporally variable snowmelt signal. The simulated initial snowmelt depletion may become more important in catchments with a rapid routing of snowmelt on frozen soils with reduced permeability (Cooper et al. 1993, Shanley and Chalmers 1999, Carey and Quinton 2004). Soil freezing has been shown to promote snowmelt runoff on wetland-covered areas at the Krycklan site (Laudon et al. 2007). The influence of soil freeze/thaw on runoff is not presently parameterised in the STARR model, which could be further developed. Future work should investigate if the simulated process of temporally progressing snowmelt enrichment becomes important in catchments with a known strong permafrost or seasonal freeze/thaw influence on early runoff.In our parsimonious snow isotope simulations we assume full isotope mixing in the snowpack (Eq. 1) at each daily time step, which is known to conflict field observations showing that snowpacks typically maintain a layered structure through the winter (Rodhe 1981, Dahlke and Lyon 2013). Furthermore, snow sublimation and melt fractionation primarily take place in the top snow layers, and are not likely instantaneously mixed in the snowpack (Claassen and Downey 1995, Evans et al. 2016), whereas we assume fractionation with respect to the bulk snowpack. However, the error caused by the full-mixing assumption is reduced by the fact that snowpack is typically homogenised during snowmelt when diurnal melt/refreeze processes take place in the snowpack (Taylor et al. 2001, Unnikrishna et al. 2002, Koeniger et al. 2008). The majority of snowpack outflow is generated during the overall snowmelt when isotopes in the snowpack are subjected to mixing, which gives empirical ground to our simplification. The limitations of the snow isotope modelling regarding the full-mixing assumption and potential biases caused by rain-on-snow events and blowing wind redistribution are further discussed in parallel work by Ala-aho et al. (2017b). In that study we also provide further evidence for the usefulness snow isotope modelling approach by finding a good agreement between simulated snowmelt isotopes and snowmelt lysimeter data sampled in Bogus and Krycklan. With the present study we demonstrate that even with the relatively simple isotope model we are able to produce improved estimates of spatially distributed snowmelt isotopes, which is called for in tracer-aided modelling of sparsely monitored snow-influenced regions (Smith et al. 2016, Delavau et al. 2017). Furthermore, we show that the stream isotopes can be used to inform parameter the snow routine through calibration, in particular for the snow sublimation fractionation.

**5 Conclusions**

Tracer-aided modelling is a powerful tool to study runoff generation processes within a catchment and, as we showed here, in inter-catchment comparison. The spatially distributed STARR modelling framework allowed us to track water age within our experimental catchments in space and time, which gave additional means to analyse the spatially distributed catchment response within and between catchments. It is well known that correctly representing snow processes in northern catchments is important in hydrological modelling and will be ever increasingly crucial with a warming climate and following changes in snow conditions. In this study, we made the first attempt to link spatially and temporally variable isotope fractionation processes in seasonal snowpacks with a tracer-aided hydrological model. Using long term datasets for three contrasting northern headwaters, we showed the importance of capturing not only the snow accumulation and melt, but also the isotopic composition of snowmelt to reproduce the streamflow isotope ratios. High frequency isotope datasets from the Bruntland and Krycklan experimental catchments were invaluable to produce a proof of concept for our modelling method in snow influenced catchments, but also coarser tracer data in the Bogus catchment were useful in constraining our modelling effort.

**Copyright statement**

The authors agree to the licence and copyright agreement by Copernicus Publications on behalf of the European Geosciences Union (EGU) in the journal Hydrology and Earth System Sciences and its discussion forum Hydrology and Earth System Sciences Discussions.

**Code availability**

The model codes are available upon request

**Data availability**

The data is available upon request

**Appendix 1**

| Snow module (for parameters in the model calibration, full description in Ala-aho et al. (2017b)) | | |
|---|---|---|
| Thermal quality of precipitation | $$P_q = \begin{array}{ll} 1 & T_a < TT_{low} \\ \frac{TT_{high}-T_a}{TT_{high}-TT_{low}} & TT_{low} < T_a < TT_{high} \\ 0 & T_a > TT_{high} \end{array}$$ <br> (A1) | $T_a$ = air temperature, <br> $TT_{low}$ = threshold temperature below which all precipitation is snow <br> $TT_{high}$ = is the threshold temperature above which all precipitation is liquid |
| Correction for snow undercatch | $$c_{snow} = \begin{array}{ll} \left(\frac{\exp(4.606-0.036*W_{can}^{1.75})}{100}\right)^{-1} & T_a < 0 \\ \left(\frac{101.04-5.62*W_{can}}{100}\right)^{-1} & T_a > 0 \end{array}$$ <br> (A2) <br><br> $$c_{snow} = \begin{array}{ll} c_{snow} + c_{corr} & c_{snow} < 1.5 \\ c_{snow} & c_{snow} > 1.5 \end{array} \quad \text{(A3)}$$ | $W_{can}$ = wind speed under canopy <br> $c_{corr}$ = empirical addition correction for snowfall |
| Rainfall phase separation | $P_{liq} = P * (1 - P_q)$        (A4) <br> $P_{ice} = P * P_q * c_{snow}$     (A5) | P = measured precipitation |
| Reduction of snow albedo | $a = 0.94^{d_{ns}^{0.58}}$       (A6) <br> $a = a^{a_{pow}}$           (A7) | $d_{ns}$ = days without snowfall <br> $a_{pow}$ = coefficient to reduce snow albedo in snowpacks older than 100 days |
| Soil module | | |
| Evaporation soil | $ET = PET * \min\left(\frac{SM}{FC*LP}, 1\right)$     (A8) <br><br> $\boldsymbol{FC = f_{cap} * S_d}$             (A9) | $PET$ = Potential evaporation <br> $SM$ = Soil moisture <br> $LP$ = Fraction of limiting actual evaporation <br> $FC$ = field capacity <br> $f_{cap}$ = **is the volumetric field capacity** <br> $S_d$ = **is the soil depth** |

| | | $\theta$ = soil porosity |
|---|---|---|
| Recharge from soil to the groundwater | $Seep = \boldsymbol{\beta_{seep}} \left(\frac{SM}{FC}\right)^{\beta_{pow}}$     (A10)<br><br>When SM > FC:<br>$Seepage = MIN(SM - FC, Seepage)$ (A11) | $\boldsymbol{\beta_{seep}}$ = **recession coefficient to determine soil recharge into groundwater**<br>$\beta_{pow}$ = power coefficient |
| Soil discharge | $Q_{soil} = k_s \left(\frac{SM}{FC}\right)^{k_{pow}}$     (A12) | $k_s$ = Recession coefficient to determine outflow from soil storage<br>$\boldsymbol{k_{pow}}$ = **Power coeffient** |
| Direct runoff | $Q_{storm} = \max(SM - \boldsymbol{SM_{max}}, 0$     (A13)<br>$\boldsymbol{SM_{max}} = \boldsymbol{\theta \cdot S_d}$     (A14) | $\mathbf{SM_{max}}$ = maximum soil storage; |
| Capillary Flux from groundwater | $Cap = C_{flux} * \left(\frac{FC-SM}{FC}\right)$     (A15) | $C_{flux}$ = Parameter for maximum capillary flux |
| Soil moisture store | $SM(t) = SM(t-1) + P_{eff} - ET - Q_{soil} - Q_{storm} - Seep + Cap$     (A16) | $P_{eff}$ = effective precipitation (sum throughfall, stemflow and snowmelt) |
| **Groundwater module** | | |
| Groundwater discharge | $Q_{gw} = k_g GW$     (A17) | $k_g$ = Recession coefficient baseflow<br>GW = groundwater store |
| Lateral groundwater flow | $Q_{lf} = k_{sat} slope(DEM + GW)$     (A18) | $k_{sat}$ = Saturated conductivity<br>$DEM$ = elevation difference between cells |
| Groundwater store | $GW(t) = GW(t-1) + Seep - Q_{gw} + \Delta Q_{lf} - Cap$ (A19) | $\Delta Q_{lf}$ = net lateral flow |
| **Routing** | | |
| Total discharge | $Q_{tot,cell} = Q_{storm} + Q_{soil} + Q_{gw}$     (A20)<br>$Q_{tot} = accutraveltimeflux(ldd, Q_{tot}, velocity)$ (A21) | $ldd$ = map with local drainage direction<br>accutraveltimeflux = routing function in PCRaster[1] |
| **Isotopes ratios** | | |

| Isotopes ratio soil (i$_s$) | $\frac{di_s(SM+SM_{pas})}{dt} = i_p P_{eff} - i_{soil} Q_{storm} - i_{soil} Q_{soil} - i_{soil} ET - i_{soil} Seep + i_{gw} Cap$ (A22) | $i_p$ = isotopes ratios effective precipitation <br> $i_{soil}$ = isotope ratios in soil storage <br> $i_{gw}$ = isotope ratios in groundwater storage <br> $SM_{pas}$ = passive storage component |
|---|---|---|
| Isotopes ratio groundwater (i$_{gw}$) | $\frac{di_{gw}(GW)}{dt} = i_{soil} Seep - i_{gw} Cap - i_{gw} Q_{gw} - i_{gw} Q_{lf,out} + i_{gw,up} Q_{lf,in}$ (A23) | $i_{gw,up}$ = isotopes ratios inflow lateral groundwater flow |
| **Water ages** | | |
| Water age soil store (Age$_{SM}$) | $\frac{dAge_{SM}(SM+SM_{pas})}{dt} = Age_P P_{eff} - Age_{SM} * Q_{soil} - 1 * Q_{storm} - Age_{SM} * ET - Age_{SM} * Seep + Age_{GW} * Cap$ (A24) | $Age_p$ = age of the precipitation (for rain equal to 1) |
| Water age groundwater (Age$_{GW}$) | $\frac{dAge_{GW}(GW)}{dt} = Age_{SM} * Seep - Age_{GW} * Cap - Age_{GW} * Q_{gw} - Age_{GW} * Q_{lf,out} + Age_{GW,up} * Q_{lf,in}$ (A25) | |

**Supplementary material**

The manuscript has animations as supplementary material

**Author contribution**

Ala-aho adapted a model developed in earlier work led by Soulsby and Tetzlaff and carried out the modelling. Data used for
15  model calibration and testing was collected by Laudon, McNamara, Soulsby and Tetzlaff for the study sites. All authors were involved in the data and model interpretation. Ala-aho prepared the manuscript with contributions from all co-authors.

**Competing interests**

The authors declare that they have no conflict of interest.

**Acknowledgements**

This work was funded by the NERC/JPI SIWA project (NE/M019896/1) and the European Research Council ERC (project GA 335910 VeWa). Numerical simulations were performed using the Maxwell High Performance Computing Cluster of the University of Aberdeen IT Service, provided by Dell Inc. and supported by Alces Software. The isotope work in Krycklan is funded by KAW Branch-Point project together with SKB and SITES. We would like to thank Marjolein van Huijgevoort for her help with the STARR code, and Masaki Hayashi and two anonymous reviewers for their insightful suggestions that significantly improved the manuscript.

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

Table 1. Calibrated parameters and parameter ranges. For Bruntland, some of the soil parameters were split into hillslopes (h) and valley bottom (v), for Krycklan between forest (f) and mire (m). For parameters which were sampled over orders of magnitudes, the parameter was log transformed in the sampling process, which is indicated by (log).

| Parameter | Eq. | Min | Max |
|---|---|---|---|
| SNOW | | | |
| $c_{corr}$ [-] | A3 | 0 | 0.3 |
| $TT_{low}$ [$^0$C] | A1 | -2 | 0 |
| $TT_{high}$ [$^0$C] | A1 | 0 | 2 |
| $a_{pow}$ [-] | A5 | 1 | 3 |
| ISOTOPES | | | |
| $E_{frac}$ [‰] | 2 | 15 | 0 |
| $M_{frac}$ [‰] | 3 | 3.5 | 0 |
| $SM_{pas}$ [mm] | A20 | 50 | 300 |
| SOIL and GROUNDWATER | | | |
| $S_d$ [m] | A7 | | |
| *Bogus* | | map, not varied | |
| *Krycklan* | | f:0.5 m:0.2 | f:1.0 m:0.5 |
| *Bruntland* | | v:0.1 h:0.5 | v:0.5, h:1 |
| $\beta_{seep}$ [-]  *Bogus only* | A8 | 1 | 10 |
| $\beta_{pow}$ [-]  *Bogus* | A8 | 1 | 3 |
| $k_s$ [day$^{-1}$] | A10 | | |
| *Bogus (log)* | | 1 | 50 |
| *Krycklan* | | f:5, m:5 | f:20, m:20 |
| *Bruntland* | | v:5 ,h:5 | v:50, h:50 |
| $k_{pow}$ [-] | A10 | | |
| *Bogus and Krycklan* | | 1 | 3 |
| *Bruntland* | | 1 | 4 |
| $k_g$ (log) | A15 | 1E-5 | 1E-3 |

Table 2. Simulated catchment average storages in groundwater storage, soil storage (water) and passive soil storage (isotope mixing) for experimental catchments in the 100 behavioural runs. Range of storage is show as 0.25 – 0.75 percentile of medians to demonstrate typical ranges in behavioural simulations.

| Site | Krycklan | Bruntland | Bogus |
|---|---|---|---|
| GW storage median [mm] range [mm] | 63 (36-128) | 540 (450 – 607) | 2182 (1791-2570) |
| Soil storage median [mm] range [mm] | 31 (25-36) | 61 (54-66) | 42 (35-50) |
| Passive storage median [mm] range [mm] | 257 (229-276) | 261 (216-287) | 186 (122-236) |

**Figures**

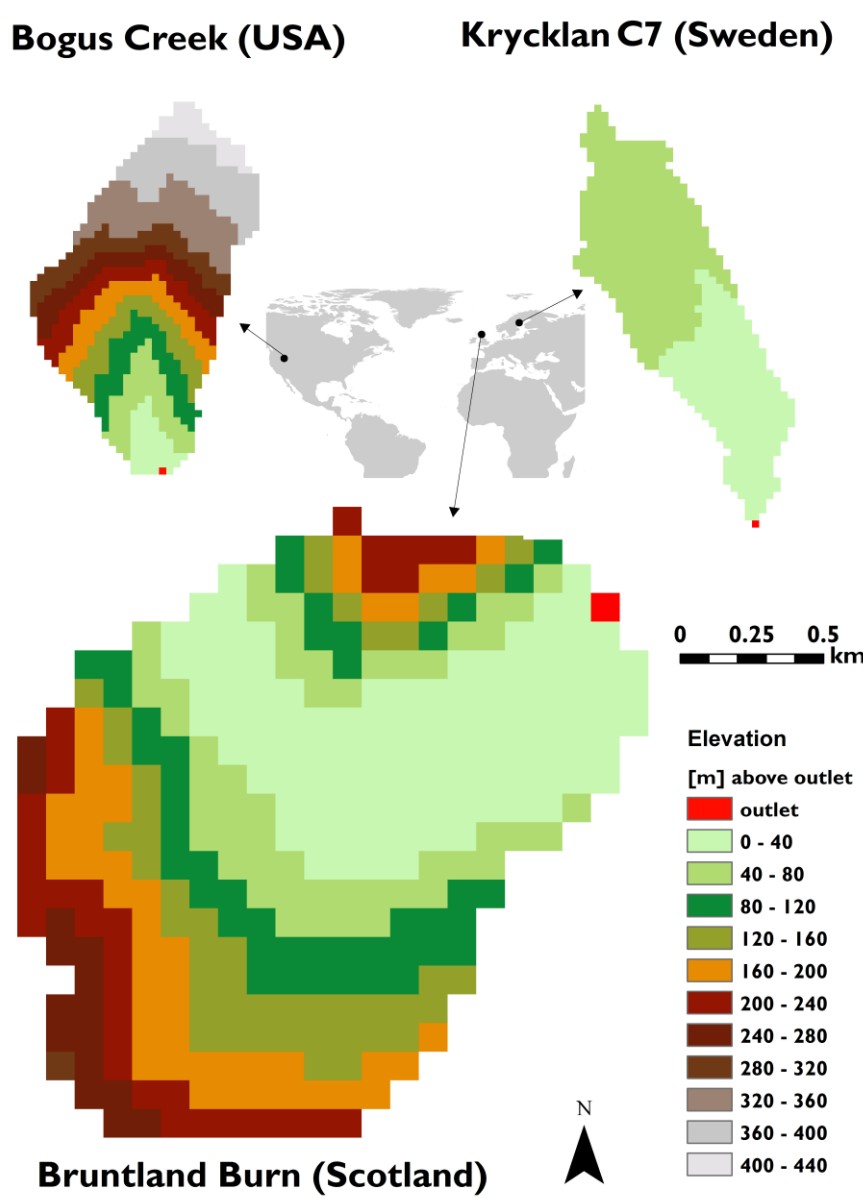

**Bogus Creek (USA)**

**Krycklan C7 (Sweden)**

**Elevation**

**[m] above outlet**

- outlet
- 0 - 40
- 40 - 80
- 80 - 120
- 120 - 160
- 160 - 200
- 200 - 240
- 240 - 280
- 280 - 320
- 320 - 360
- 360 - 400
- 400 - 440

**Bruntland Burn (Scotland)**

N

0    0.25    0.5
km

**Fig. 1. Study sites in same scale for size and colour scheme for altitude, and location of the sites. Pixel size in the figure equals the cell size of the model (25 m for Bogus and Krycklan, 100 m for Bruntland).**

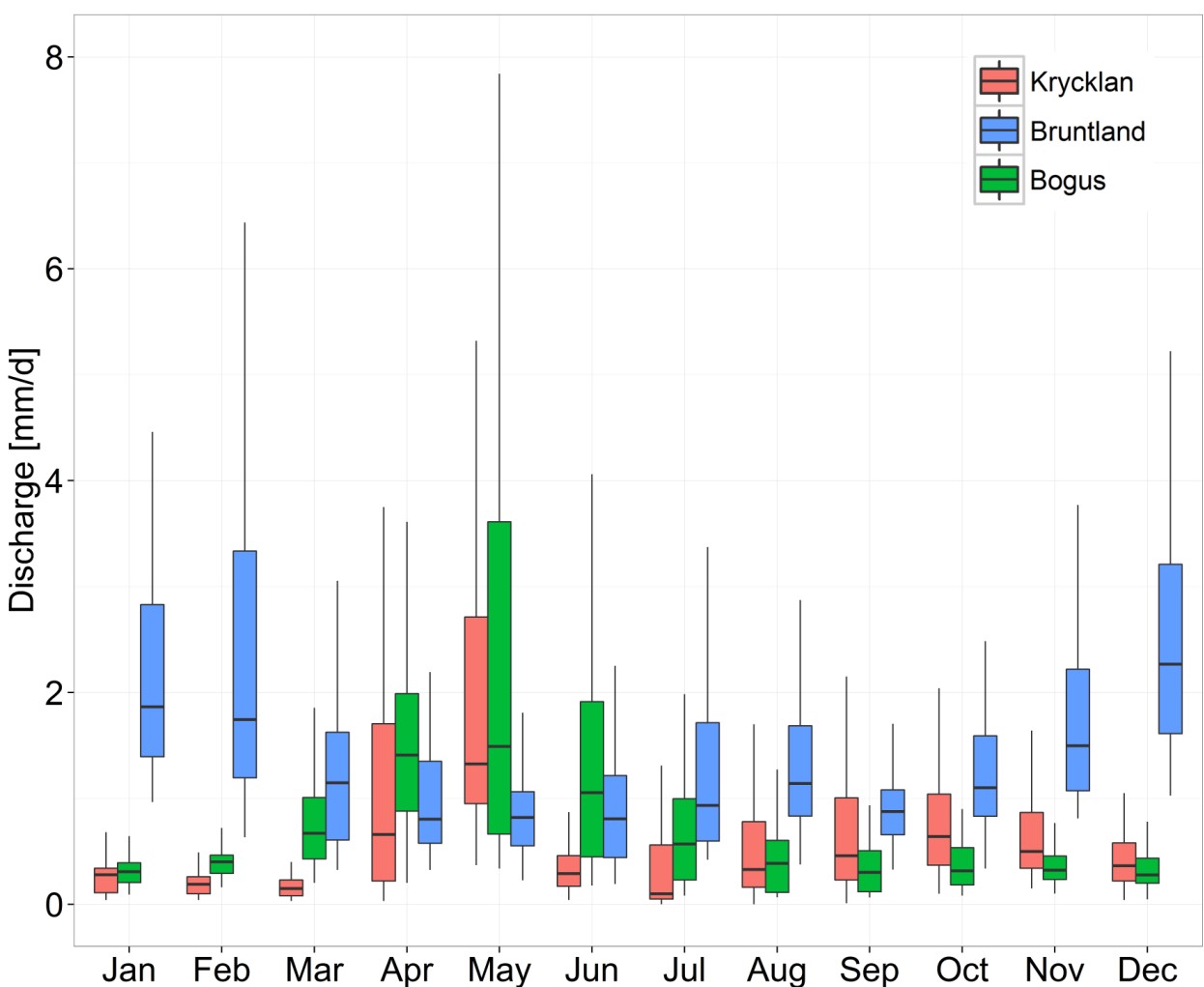

**Fig. 2. Boxplots showing the seasonality of streamflow for each catchment by grouping daily streamflow (5 years of data for Bruntland, 8 for Bogus and 10 for Krycklan) to monthly bins. Coloured box shows the 25th and 75th quantiles, small horizontal line the median and vertical line the extent of 5th and 95th percentiles. High flows for Krycklan and Bogus are brought about by snowmelt occurring typically in April-May. For Bruntland flows are more evenly distributed through the year with highest flow during December-February with occasional snowmelt influence. Colour coding: red for Krycklan, blue for Bruntland, green for Bogus are maintained through the manuscript.**

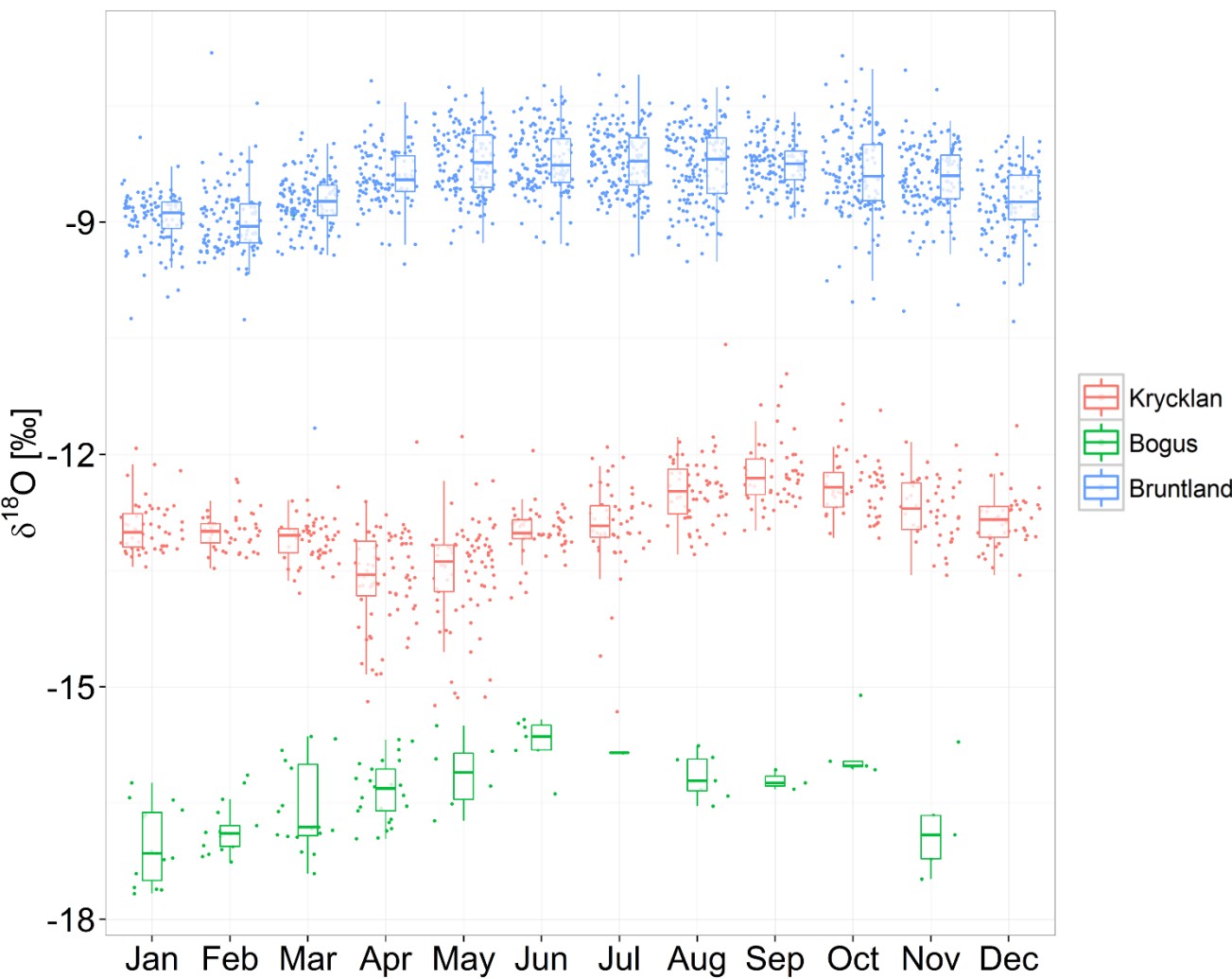

**Fig. 3. Isotope samples of streamflow for each site grouped to monthly bins where the jitter plots show all data points and boxplots summarize the span of the data. For Bogus we show data presented in (Kormos 2005) which was not used in model calibration due to problems with the flow data.**

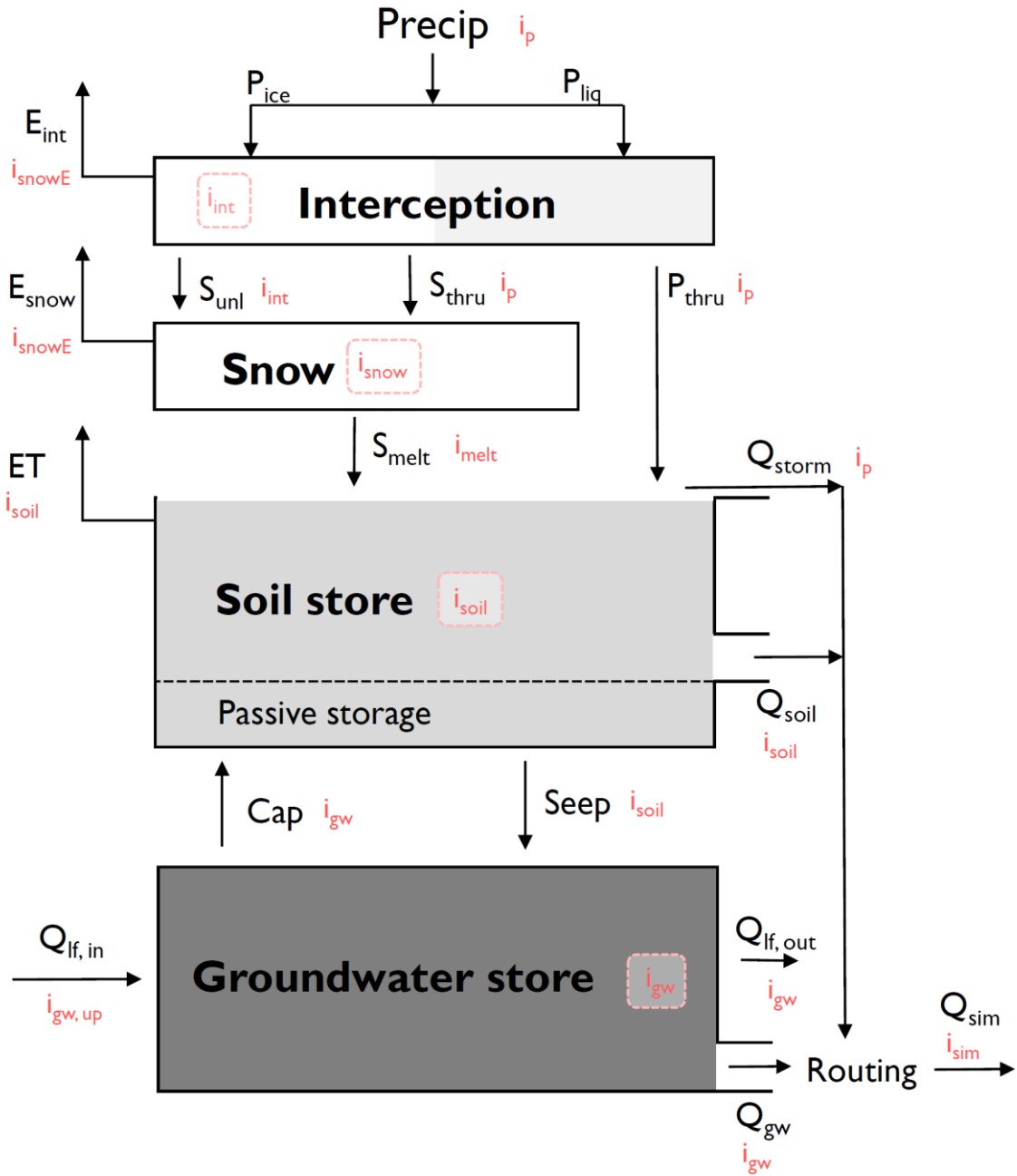

**Fig. 4. Schematic model structure of each model cell in STARR. Black arrows and boxes refer to water fluxes/storages, red i indicate the isotopic ratio of the given flux/storage. Calculation of variables and storages are presented in Appendix (A1).**

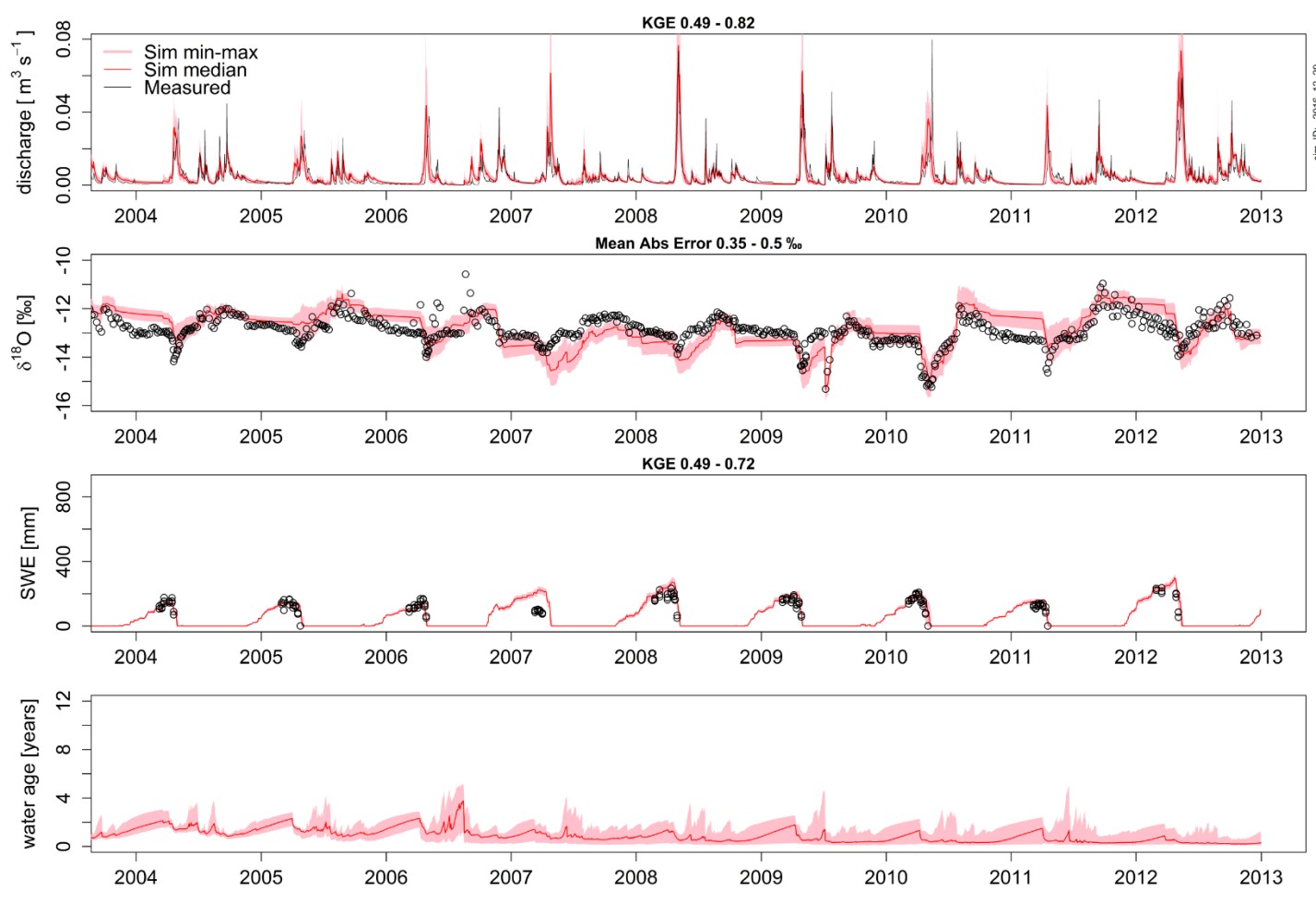

**Fig. 5.** Krycklan C7 simulation output: top panel shows the streamflow, second panel the isotope ratio at the stream outlet, third panel the snow water equivalent and lowest the simulated stream water age. GOF measures are shown on plot titles.

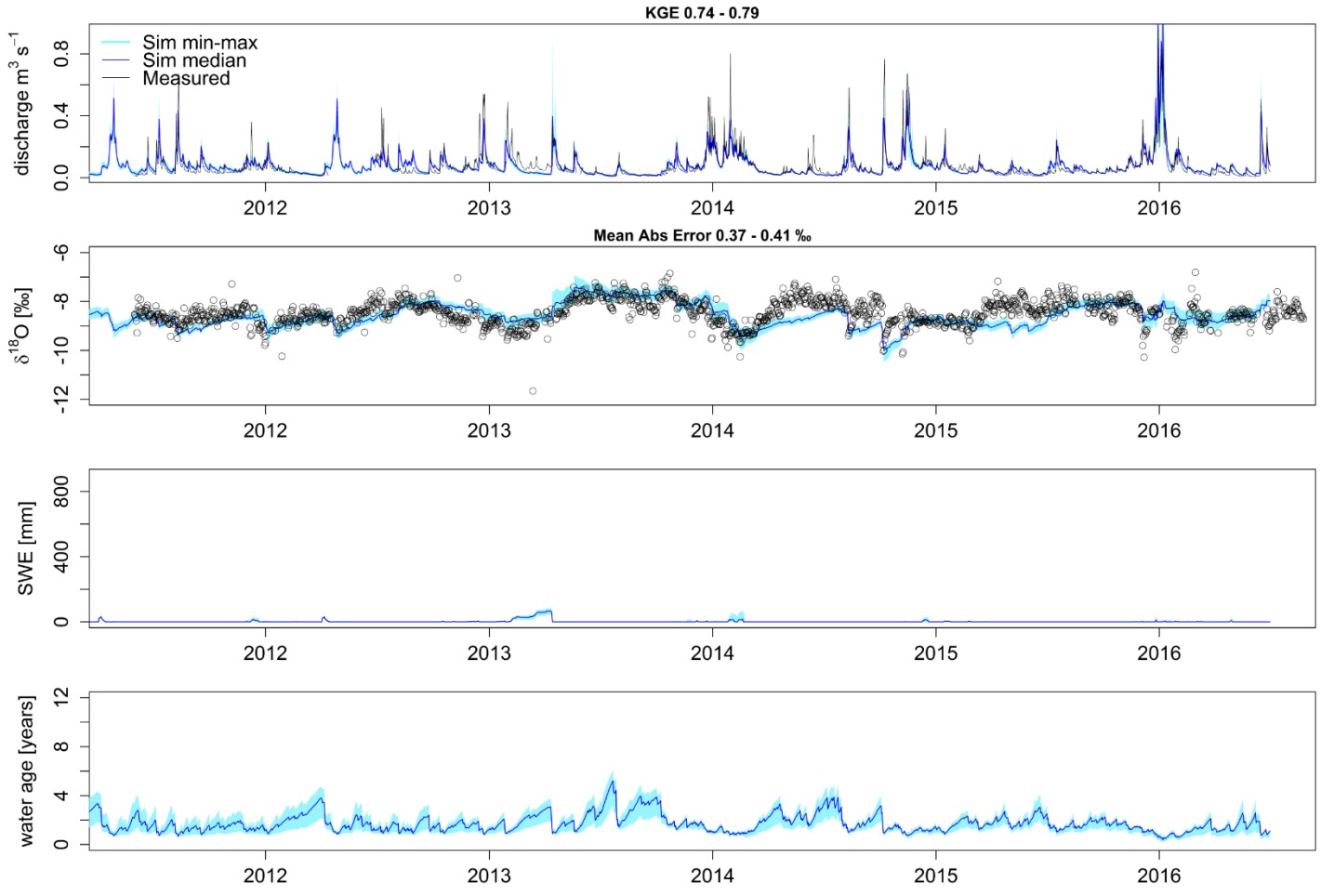

**Fig 6. Bruntland Burn simulation output: top panel shows the streamflow, second panel the isotope ratio at the stream outlet, third panel the snow water equivalent and lowest the simulated stream water age. GOF measures are shown on plot titles.**

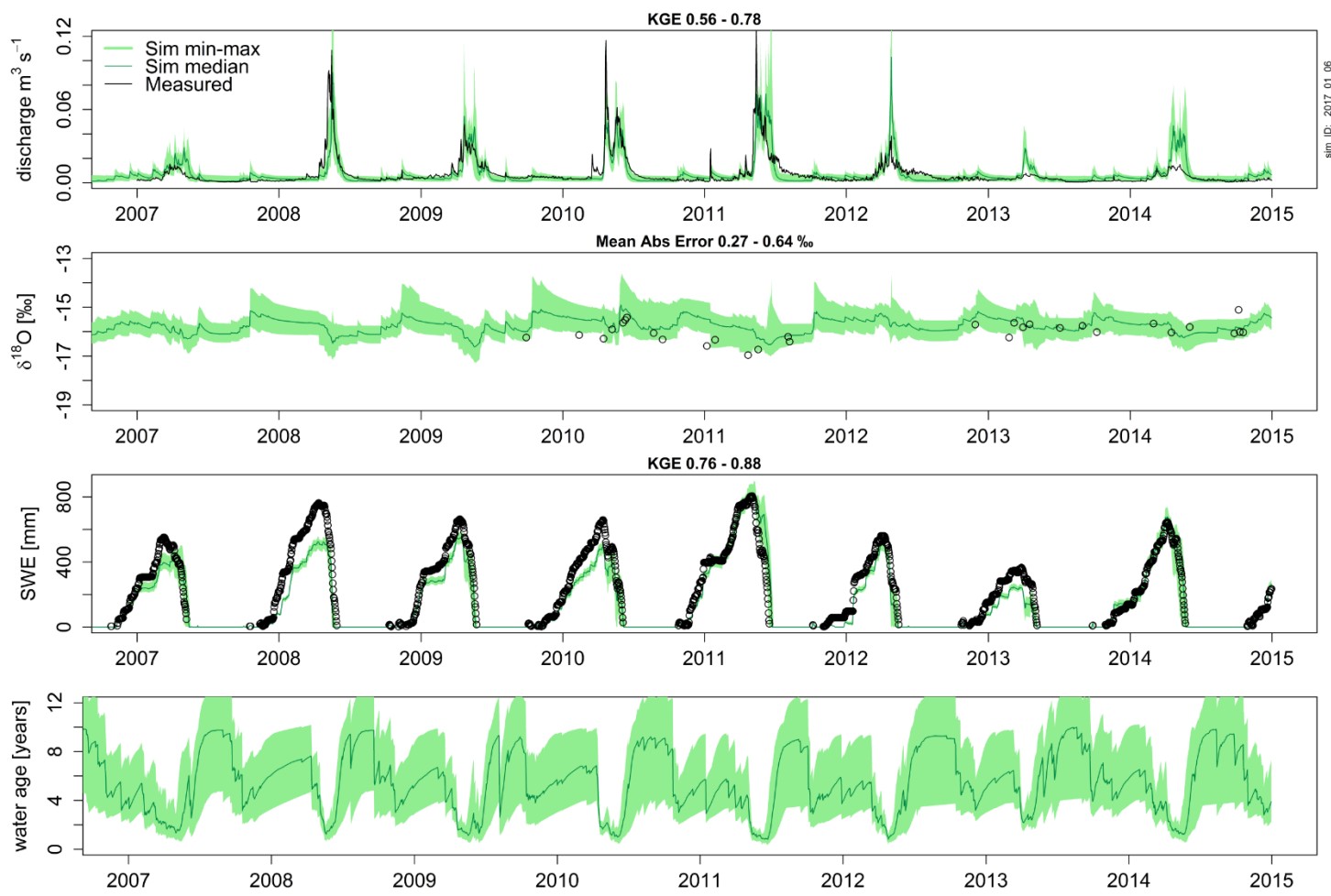

**Fig. 7.** Bogus Creek simulation output: top panel shows the streamflow, second panel the isotope ratio at the stream outlet, third panel the snow water equivalent and lowest the simulated stream water age. GOF measures are shown on plot titles.

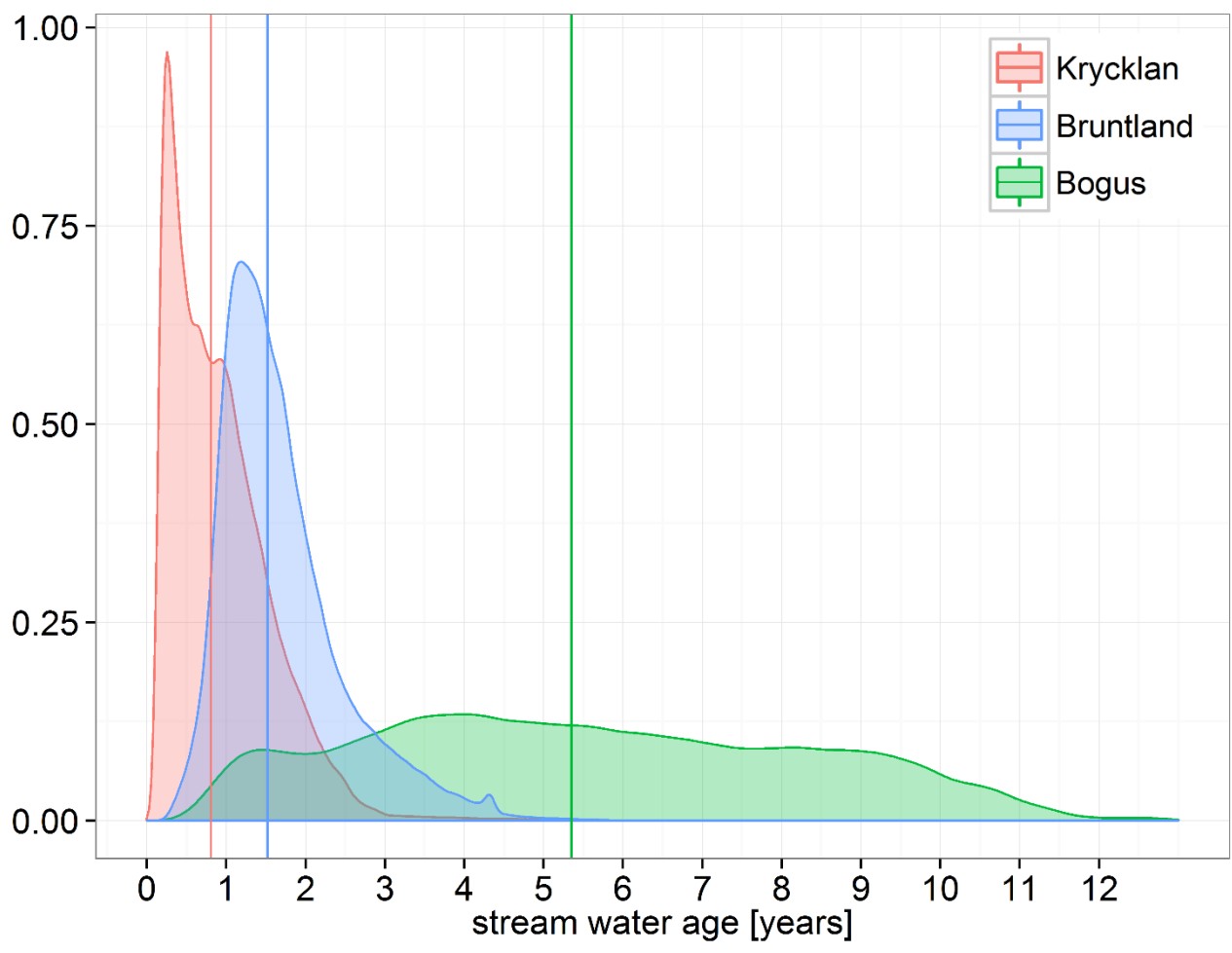

**Fig. 8. Probability density functions of the simulated water ages for the best 100 runs for all three catchments. The vertical lines mark the median age in each catchment.**

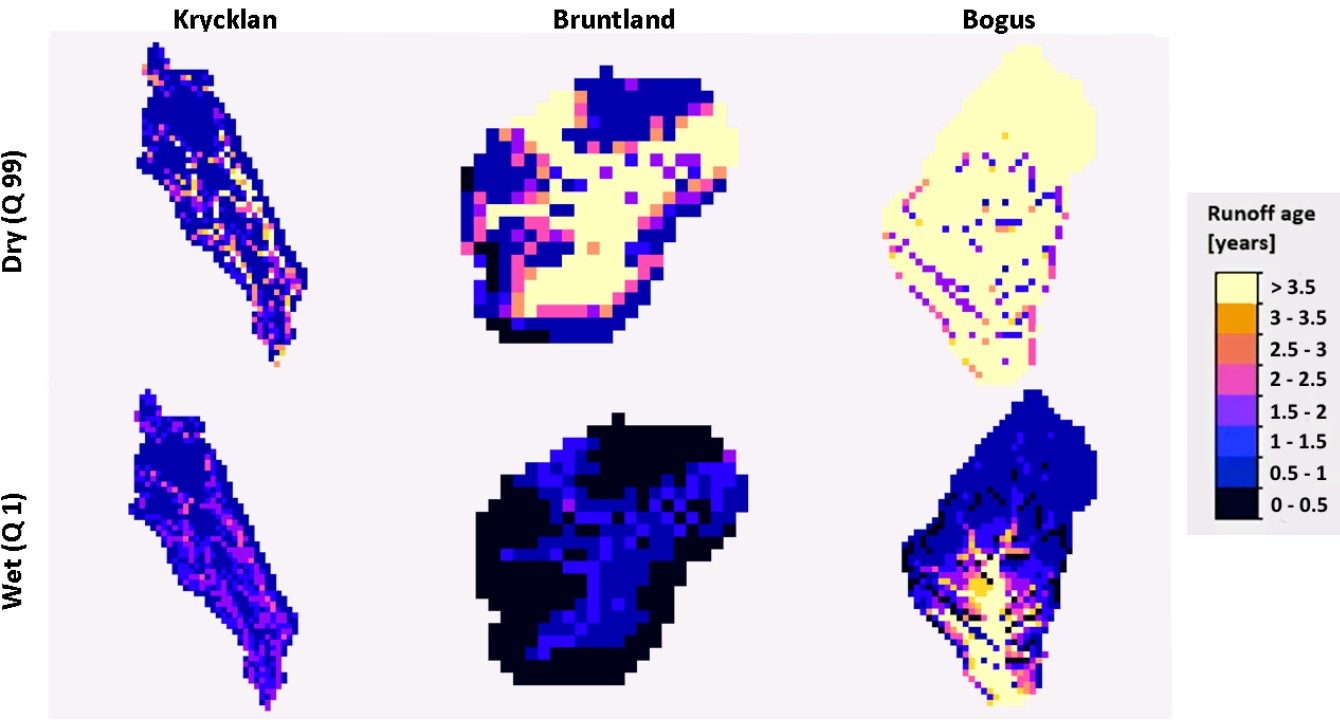

**Fig. 9. Spatially distributed water age in each catchment for dry (1% quantile) and wet (99% quantile) conditions. Model outputs are extracted from a single model run with the "best" model fit in each site.**

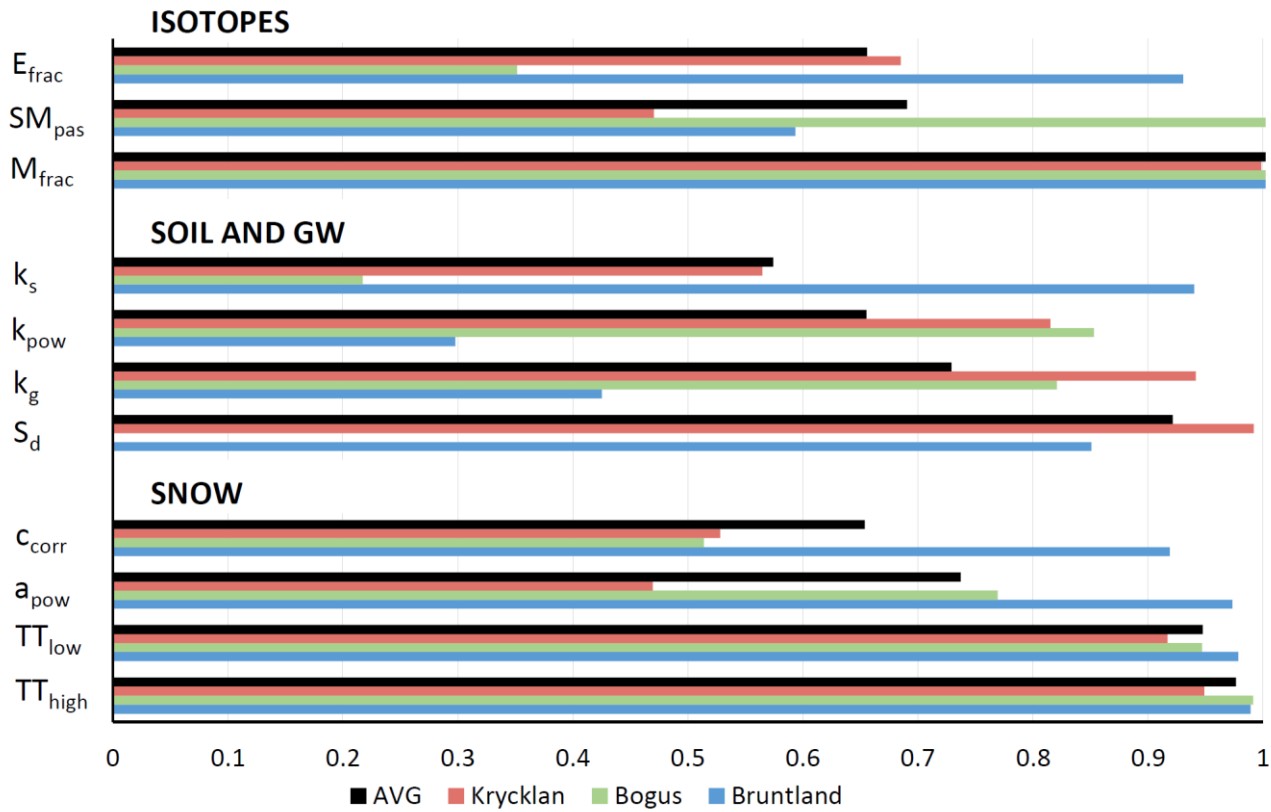

**Fig. 10.** Sensitivity of model parameters shown as ratio of pre and post calibration standard deviation – smaller the ratio the more constrained the post-calibration parameter interpreted as parameter sensitivity. The parameters are grouped by model modules and organised within modules from most sensitive to least sensitive based on average for all experimental catchments.