# Peer review of "Using isotopes to constrain water flux and age estimates in snow-influenced catchments using the STARR (Spatially distributed Tracer-Aided Rainfall-Runoff) model"

_Hydrology and Earth System Sciences, 2017_

## Referee Comment (RC1) · Anonymous Referee #1 · 17 Apr 2017

The paper presented a new study on specifically the stream water oxygen isotope by spatially distributed STARR model coupling with the snow evaporation fractionation and snow melting fractionation at three northern northern catchments with different annual precipitation and winter snow accumulation. The improved simulation work captured pretty well the observed seasonal stream water oxygen isotope variations at two of the catchments. The study also demonstrated the importance of snow evaporation and melting in the adjusting the temporal variations of steam water isotope. This work has the potential of wide applications in isotope hydrology in other catchments with significant snowpack in winter season. 1. A comparison between local precipitation and river

water $\delta 18O$ may help to see the the impact of precipitaiton on river water$\delta 18O$. And I wonder if we can see the lag between precipitaion $\delta 18O$ and river water $\delta 18O$, and this lage is related to the age of water? 2. Isotope fractionation in the surface evapotranspiration should be introduced in the paper, even it is included in the previous publications, since it is another process significantly affect the stream water isotope. 3. The d-excess in water may more sensitive to evaporation, and therefore, provide more unequivical proof in the water cycle in snow evaporation and melting. 4. The inconsistence between the simulated stream water $\delta 18O$ and observed stream water$\delta 18O$ probablly hints the impac of underground water at Krycklan. With decreasing trend in both river discharge and stream water $\delta 18O$, there is probably a increasing ratio of deep underground water with lower water$\delta 18O$. This agree with the increasing water age. However, the underground water $\delta 18O$ data is necessary for further discussion. 5. From Figure 11 it is difficult to to see how different parametering can affect the simulated results. There are minor questions: 6. In all the text, please include the full name for the term while they are first mentioned, e.g. SWE (snow water equivalen?), DCEW, MET, SNOTEL, 7. There are dummy text in Line 25-27, Page 3ïijŽ "Suspendisse a elit ut leo pharetra cursus sed quis diam. Nullam dapibus, ante vitae congue egestas, sem ex semper orci, vel sodales sapien nibh sed lectus. Etiam vehicula lectus quis orci ultricies dapibus. In sit amet lorem egestas, pretium sem sed, tempus lorem." 8. Page 11ïijŇ Line 29, change from "different to" to "different from". 9. What is passive storeage?

---

## Referee Comment (RC2) · M. Hayashi (Referee) · 17 Apr 2017

GENERAL COMMENTS

The manuscript presents an innovative approach to examine the residence time of water in catchments by using a numerical model to simulate the flow and isotopic composition of streams draining small catchments. The model has a relatively simple construction but it captures the integrated effects of spatially distributed sources and reservoirs of water and provides a useful tool for improving our understanding of catchment hydrological processes. This is a nice piece of work and warrants publi-
cation in this journal. However, I have noted a few issues that need to be addressed before the manuscript is considered for publication. Please see my specific comments below.

SPECIFIC COMMENTS

P3, L25-28. I am not sure if these sentences (in Latin?) are meant to be here.

P5, L21-23. Were event-by-event precipitation samples available for both snow and rain? It is straight forward to collect rain samples, but I am not sure how snow samples were collected. Please explain.

P5, L26. Please spell out DCEW at its first appearance.

P5, L33. What is the elevation of Svartberg meteorological station in relation to the catchment outlet?

P6, L1. Where are these meteorological stations? Can you show them in Fig. 1?

P6, L2. How far is this station? What is the elevation?

P6, L6. What is the elevation of the SNOTEL station?

P6, L8. How were these lapse rates determined? These rates may vary between summer and winter. What is the justification for using constant values?

P6, L12. Where was SWE measured? Can you indicate the location in Fig. 1?

P7, L9. What is the difference between "snow storage" and "ground snowpack"?

P7, L10. This equation assumes instantaneous mixing of snow within the snowpack. This may be a reasonable model for a thin snowpack, but its validity is questionable for a thick snowpack typical of mountainous catchment (see SWE graph in Fig. 7). Fractionation associated with sublimation and evaporation occurs from snow near the surface, which is not easily mixed with the rest of the snowpack. Similarly, snowmelt fractionation occurs near the surface, not from the entire snowpack. I see this as
a major deficiency in the snow isotope module of this model. Its effects on model performance need to be discussed more openly and carefully. It is highly desirable to add isotope data for snowpack or snowmelt percolation collected by snow lysimeter (P8, L3) to validate the assumption.

P7, L22. It appears that the model uses a constant value for evaporation fractionation factor, whereas it is expected to vary with meteorological conditions such as relative humidity and wind speed. Please present justification for using a constant value.

P7, L20. Snow loss to the atmosphere occurs by two different processes depending on snow surface temperature; sublimation under 0 C, and evaporation of melt water at 0 C. Resulting isotopic fractionation factors may be different. This is a subtle point, but should be discussed.

P7, L26. What is the reasoning for dividing M_frac by d_melt? Is this purely empirical or does it have a theoretical basis?

P8, L1. I am generally against citing unaccepted "in review" manuscript because there is no guarantee that the manuscript will be published. Please avoid using the in review manuscript, or at least include it in the reference list with the journal name.

P8, L3-4. What kind of algorithm is used for the new snow module? Is it still a degree-day model? I note that radiation is included in the data set (P5, L32)? Please include a brief explanation of the model. Note that in a catchment with rugged topography such as Bogus, slope aspect and angle may have a strong influence on the spatial distribution of snow accumulation and melt.

P8, L6. This is another questionable assumption. Snow accumulates from the bottom to the top during winter without much mixing. Snow melt and evaporation occurs from the top. Therefore, it is questionable to assume complete mixing for water age. Please point this out in texts and discuss potential errors resulting from this assumption.

P8, L8-10. I do not quite understand this sentence. How is FC defined? By calibration?

[Figure]

It is not listed in Table 1.

P8, L11. The need for adaptation became apparent. How?

P8, L13. Depending on the soil thickness, root density, and other complex factors, it is unlikely that evaporation age is equal to average soil water age. Please discuss this issue carefully.

P8, L16. There appear to be 13 parameters listed in Table 1. If three values are used for each of the 13 parameters, there are roughly 1.6 million combinations (3^13) of parameter values. How were the 10,000 combinations selected? Please explain.

P8, L20. Were empirical coefficients used in the snow module (e.g. coefficient for degree-day model)? If so, how were they calibrated? Please explain.

P8, L25. Is this field capacity the same FC as the one described in L8-10 above?

P9, L17. The standard metric for stream flow calibration is Nash-Sutcliff efficiency (NSE), which is familiar to most of the readers of this journal. I do not see a strong justification for using Kling-Gupta efficiency (KGE). I suggest NSE be used instead of KGE. If not, please present a stronger justification for using KGE and include its definition (e.g. equation).

P9, L28 – P10, L3. I read this section several times, and still could not understand the definition of F ( ) and how it was constructed. Please re-write the section more clearly.

P10, L6-9. This section was also very difficult to follow. Please re-write.

P10, L30-31. The model is calibrated for stream isotopic composition, but it is not clear how well the isotopic compositions in groundwater and soil water are simulated. Laudon et al. (2013) describes systematic sampling programs for soil water and groundwater at Krycklan catchment. Please include a comparison of simulated and observed soil water and groundwater data, where available.

P15, L2. It is true that STARR has a spatially distributed model structure, but it uses

uniform values of model parameters for an entire catchment. In reality, the spatial distribution of material properties (e.g. soil type and thickness, bedrock type, vegetation) has an important effect on catchment hydrological responses. That is why many models use hydrological response unit (HRU) approaches to capture catchment heterogeneity. I suggest that the authors discuss the lack of heterogeneity in the current configuration of STARR and its potential implication for model performance and estimated water age.

P16, L10. High sensitivity of E_frac in Fig. 10. This is a bit misleading because the highest sensitivity is observed for Bogus catchment, which had few data points. Fig. 10 indicates low sensitivity for E_frac for Bruntland catchment. If the authors want to showcase the snow isotope model as one of the highlights of this work, then this topic needs to be explored a bit more carefully. Please note my comments on the isotopic fractionation in P7, L10.

P17, L8. I do not think isotope data from Bogus catchment had exceptionally high quality. Please clarify.

P17, L11. I do not think the need to incorporate isotopic evaporative fractionation is convincingly demonstrated. Please see my comment on Fig. 10 above.

Fig. 5. It is impossible to see the difference between red line and pink band in the top figure. Can you use different color combination (e.g. blue and pink) and use the same color for all of Fig. 5, 6, and 7? I do not see a real need for using different colors for different catchments.

Fig. 9. Please include a scale and a north arrow for each catchment.

---

## Referee Comment (RC3) · Anonymous Referee #3 · 9 May 2017

The article developed a calculation scheme to describe the spatially and temporally variable isotope fractionation processes in seasonal snowpacks, and the calculation scheme was linked to the STARR model to simulate the stream flows, isotope ratios and snow pack dynamics in three long term experimental catchments. General Comments: 1. The spatially distributed water age was simulated as shown in Fig.9. Does the spatially distributed water age is estimated by averaging the water age of surface water, soil stored water and groundwater stored water in each computing unit? The water age of incoming precipitation is taken as 1, which means the water age of stored water in the catchment should be larger than 1. Why the calculated water age is less

than 1, even equal to 0, in many places of the three catchments? Specific Comments: 1. L18P1, P4 It is better to specify how long does the "exceptionally long" mean? The observation period for the hydrometric data and the stable water isotopes date should also be given in the "Study sites" section for better understanding the simulation process and calibration results. 2.L24P1, The geology setting is only briefly described for the Bogus Creek, but not given for the other two study sites in the article. How to make a conclusion that the geology is an important factor causing contrasting water age distribution among the three catchments? 3. L8P2, The meaning of abbreviation of "STARR" has been given in L2P2. 4. L12P2, The literature "Ala-aho et al., in review" has been cited several times in the article. It's better to give a specific published journal or publishing house etc. of the literature. 5. L25P2, The sentence seems written in Latin. 6. L26P5, The meaning of DCEW should be given here. 7. L33P5-L5P6, The meteorological date from stations in the neighboring catchments are used for the Krycklan and Bruntland Burn catchments. It's better to describe the distance of the stations from the study sites, or give the latitude and longitude coordinates of the stations for better judging the reliability of the inputted meteorological data in the model. 8. L31P8, What does the "kg" means? Hydraulic conductivity?
* * *

---

## Author Comment (AC1) · 1 Jun 2017

**Author response to Referee #1 comments**

Manuscript hess-2017-106, "Using isotopes to constrain water flux and age estimates in snow-influenced catchments using the STARR (Spatially distributed Tracer-Aided Rainfall-Runoff) model" by Ala-aho, P. et al.

We are grateful for the comments by anonymous referee #1 on our manuscript and referee's constructive suggestions for improvement in revision. We have now carefully addressed the comments suggest corrections, clarifications and deeper discussion requested. We hereby provide our point by point responses how the comments by referee #1 will be addressed in the revised manuscript.

Yours sincerely,

Pertti Ala-aho

**Referee #1 comments**

The paper presented a new study on specifically the stream water oxygen isotope by spatially distributed STARR model coupling with the snow evaporation fractionation and snow melting fractionation at three northern northern catchments with different annual precipitation and winter snow accumulation. The improved simulation work captured pretty well the observed seasonal stream water oxygen isotope variations at two of the catchments. The study also demonstrated the importance of snow evaporation and melting in the adjusting the temporal variations of steam water isotope. This work has the potential of wide applications in isotope hydrology in other catchments with significant snowpack in winter season.

We thank Referee #1 for her/his support of our work

1. A comparison between local precipitation and river water 18O may help to see the the impact of precipitaiton on river water 18O. And I wonder if we can see the lag between precipitaion 18O and river water 18O, and this lage is related to the age of water?

Good suggestion; prior work at the sites has used the phase lag in precipitation and stream water to estimate water transit times through convolution integral techniques. Work for Krycklan by Peralta-Tapia et al. (2016) was readily referenced in the discussion, a reference for Bruntland Burn by Hrachowitz et al. (2010) will be added.

2. Isotope fractionation in the surface evapotranspiration should be introduced in the paper, even it is included in the previous publications, since it is another process significantly affect the stream water isotope.

We agree with the comment that the missing evaporation fractionation processes can deteriorate the model performance in the summer months, which is most evident in the Bruntland Burn timeseries, as discussed in the manuscript (P14L7). We will expand this discussion by:

*"Smith et al (2016) have successfully included the soil evaporative fractionation in their spatially distributed tracer-aided simulations, and similar approaches could be adopted to the STARR model to improve model realism during summer periods with elevated evaporation."*

3. The d-excess in water may more sensitive to evaporation, and therefore, provide more unequivical proof in the water cycle in snow evaporation and melting.

Thank you for the good suggestion. Parsimonious conceptualisation of our snow isotope simulations does not differentiate between equilibrium and kinetic fractionation processes, therefore we cannot readily simulate d-excess. Furthermore, during winter when air temperature is low and relative humidity high, the equilibrium fractionation process can be expected to dominate over kinetic fractionation, which would imply minimal d-excess signal. We propose to add the following explanation and discussion in the manuscript:

*"In our parsimonious isotope simulation approach we did not differentiate between kinetic and equilibrium fractionation in snow sublimation, and we only simulated only the $\delta^{18}O$ isotope because of better data availability in all sites. This simplification prevented us from simulating additional isotopic indices for evaporation, such as the d-excess (Dansgaard 1964), that would indicate deviations from the meteoric water caused by kinetic fractionation. In typical winter conditions with low air temperature and high relative humidity, we would expect the equilibrium fractionation to dominate over kinetic fractionation (Gat and Gonfiantini 1981), therefore making the differentiation between the two processes of lesser importance."*

4. The inconsistence between the simulated stream water 18O and observed stream water18O probablly hints the impac of underground water at Krycklan. With decreasing trend in both river discharge and stream water 18O, there is probably a increasing ratio of deep underground water with lower water 18O. This agree with the increasing water age. However, the underground water 18O data is necessary for further discussion.

This is a perceptive insight to the modelling results which was not discussed in the initial manuscript. We suggest to add the following discussion on groundwater contribution on stream isotopes and water age at the Krycklan site, and discuss the wider implications of the difficulties in defining initial conditions for the groundwater storage in the MC calibration of conceptual models:

*"Another parameterisation issue in our work rises from specifying initial conditions for the groundwater storage for the Monte-Carlo runs. If the initial GW storage is not in "balance" with the magnitude of the outflow coefficient (kG), which is randomly varied in the calibration, it can lead to GW storage reduction or increase over time. Our simulations at the Krycklan site show symptoms of such imbalances between the kG parameter and the initial GW storage, as the variability and median in the simulated stream water age declines over the ten year period (Fig. 5). The non-stationarity in age suggest that the groundwater influence (GW storage has older water) reduces over time. In further analysis (data not show) in most of the behavioural simulations the total GW storage in Krycklan in fact grows smaller over time. A longer spin-up period for the Krycklan simulations would alleviate the issue, with the burden of increased runtimes. In addition, even though our simulations for streamflow during winter is well captured (Fig. 5), the isotope composition in some winters does not shift adequately towards more depleted values (isotopes in deep groundwater between -13 and -14 ‰ (Peralta-Tapia et al. 2015)), suggesting a too low groundwater contribution. The misfit in winter isotopes suggests that the model has problems in switching from soil source to a more depleted groundwater source during winter. It should be pointed out, that such analysis and insights are only possible because of the ability of the STARR model to simulate stable water isotopes and water ages – these issues would not become apparent if using only streamflow hydrograph to evaluate the model performance."*

5. From Figure 11 it is difficult to to see how different parametering can affect the simulated results.

We assume the referee is here referring to Fig. 10. This is a fair point, as the only the sensitivity of the parameter is plotted, not the range in which the parameters in the behavioural parameters calibrate in the parameter space. The purpose of the figure is to identify the most sensitive parameters and discuss why sensitivities differ between sites, and for this purpose the parameter values are in our opinion of lesser importance.

There are minor questions:

6. In all the text, please include the full name for the term while they are first mentioned, e.g. SWE (snow water equivalen?), DCEW, MET, SNOTEL,

Full names for abbreviations on their first occurrence will be added

7. There are dummy text in Line 25-27, Page 3ïïjŽ "Suspendisse a elit ut leo pharetra cursus sed quis diam. Nullam dapibus, ante vitae congue egestas, sem ex semper orci, vel sodales sapien nibh sed lectus. Etiam vehicula lectus quis orci ultricies dapibus. In sit amet lorem egestas, pretium sem sed, tempus lorem."

Apologies, this will be removed

8. Page 11ïïjˇN Line 29, change from "different to" to "different from".

Will be changed

9. What is passive storeage?

We agree that the passive storage concept was not properly explained given its importance to the model. We propose to add the following clarification:

*"Like its predecessors, the STARR model utilises a concept of passive storage in isotopic mixing in the soil (Birkel et al. 2015). Passive storage parameterises the water stored in the soil that does not relate to changes in discharge, but increases the total mixing volume of isotopes."*

List of suggested additional references:

Dansgaard, W.: Stable isotopes in precipitation, Tellus, 16, 436-468, 1964.

Gat, J. R. and Gonfiantini, R.: Stable isotope hydrology. Deuterium and oxygen-18 in the water cycle, IAEA, Vienna, 1981.

Peralta-Tapia, A., Sponseller, R. A., Ågren, A., Tetzlaff, D., Soulsby, C. and Laudon, H.: Scale-dependent groundwater contributions influence patterns of winter baseflow stream chemistry in boreal catchments, Biogeosciences, 120, 847-858, 2015.

---

## Author Comment (AC2) · 1 Jun 2017

**Author response to Referee #2 M. Hayashi comments**

Manuscript hess-2017-106, "Using isotopes to constrain water flux and age estimates in snow-influenced catchments using the STARR (Spatially distributed Tracer-Aided Rainfall-Runoff) model" by Ala-aho, P. et al.

We are grateful for the insightful comments by Prof. Masaki Hayashi (ref #2) on our manuscript and his constructive suggestions for improvement in revision. We have now carefully addressed the comments providing the corrections, clarification and deeper discussion requested. We hereby provide our point by point responses how the comments by ref #2 are addressed in the revised manuscript.

Yours sincerely,

Pertti Ala-aho

**Referee #2 comments**

GENERAL COMMENTS

The manuscript presents an innovative approach to examine the residence time of water in catchments by using a numerical model to simulate the flow and isotopic composition of streams draining small catchments. The model has a relatively simple construction but it captures the integrated effects of spatially distributed sources and reservoirs of water and provides a useful tool for improving our understanding of catchment hydrological processes. This is a nice piece of work and warrants publication in this journal. However, I have noted a few issues that need to be addressed before the manuscript is considered for publication. Please see my specific comments below.

We appreciate Professor Hayashi's positive assessment of our paper.

SPECIFIC COMMENTS

P3, L25-28. I am not sure if these sentences (in Latin?) are meant to be here.

Apologies, these will be removed

P5, L21-23. Were event-by-event precipitation samples available for both snow and rain? It is straight forward to collect rain samples, but I am not sure how snow samples were collected. Please explain.

We suggest the following explanation to be added for snow isotope sampling:

*"For Bruntland only liquid precipitation was sampled because of rarely occurring snowfall events. For Krycklan with more persistent snow cover, precipitation was sampled daily following every snow fall, melted in a cool room (+8 °C) and subsequently measured for volume using a fine graded measurement cylinder."*

P5, L26. Please spell out DCEW at its first appearance.

Explanation for DCEW will moved to its first appearance

P5, L33. What is the elevation of Svartberg meteorological station in relation to the catchment outlet?

Elevation will be added

P6, L1. Where are these meteorological stations? Can you show them in Fig. 1?

Written specifications of the meteorological station locations and their elevations with respect to the catchments will be added for all sites, which we hope will clarify their location adequately. Because they are not within the catchments for Bogus and Krycklan and the Fig. 1 presents catchments in the same figure and scale, we propose not to include the stations in the Fig. 1 to avoid confusion.

P6, L2. How far is this station? What is the elevation?

Details for station locations and elevation will be added, see the response to comment above

P6, L6. What is the elevation of the SNOTEL station?

Elevation in reference to catchment outlet will be added, see the response to comment above

P6, L8. How were these lapse rates determined? These rates may vary between summer and winter. What is the justification for using constant values?

Thank you for the perceptive comment. Lapse rates for air temperature and precipitation will explained and their uncertainties will be pointed out as follows:

*"A spatially distributed environmental lapse rate of -0.6 C/100 m was applied to air temperature measurements according to the moist adiabatic lapse rate (Goody and Yung 1995). A +5.4 %/100 m increase in precipitation was measured in the Bruntland along a hillslope covering 200 m elevation difference, and the parameter value was transferred to Bogus. We used temporally constant lapse rates, but they may vary in different seasons, latitudes, and orographic influences (Stone and Carlson 1979, Sevruk and Mieglitz 2002). Altitude effects are negligible for the gently sloping Krycklan site (Karlsen et al. 2016)."*

P6, L12. Where was SWE measured? Can you indicate the location in Fig. 1?

We will add a sentence to specify the location because it cannot be easily incorporated into Fig. 1

P7, L9. What is the difference between "snow storage" and "ground snowpack"?

Snow storage is present on both canopy (interception) and ground, and in both storages we assume full mixing. This will be clarified in the manuscript:

*"Isotopes in the snow storage (ground snowpack and interception storage) are fully mixed within each time step."*

P7, L10. This equation assumes instantaneous mixing of snow within the snowpack. This may be a reasonable model for a thin snowpack, but its validity is questionable for a thick snowpack typical of mountainous catchment (see SWE graph in Fig. 7). Fractionation associated with sublimation and evaporation occurs from snow near the surface, which is not easily mixed with the rest of the

snowpack. Similarly, snowmelt fractionation occurs near the surface, not from the entire snowpack. I see this as a major deficiency in the snow isotope module of this model. Its effects on model performance need to be discussed more openly and carefully. It is highly desirable to add isotope data for snowpack or snowmelt percolation collected by snow lysimeter (P8, L3) to validate the assumption.

The perceptive comment shows the reviewers knowledge in the field, and we agree that the mixing and fractionation assumptions in the snow isotope model is a major and justifiably challenged model assumption. However, for our purposes, the focus is not the accurate simulation of the snow pack evolution, but the ability to predict the pulse of depleted melt water. In this sense, the assumption is appropriate. We propose to add the following for justification in the discussion:

*"In our parsimonious snow isotope simulations we assume full isotope mixing in the snowpack (Eq. 1) at each daily time step, which is known to conflict field observations showing that snowpacks typically maintain a layered structure through the winter (Rodhe 1981, Dahlke and Lyon 2013). Furthermore, snow sublimation and melt fractionation primarily take place in the top snow layers, and are not likely instantaneously mixed in the snowpack (Claassen and Downey 1995, Evans et al. 2016), whereas we assume fractionation with respect to the bulk snowpack. However, the error caused by the full-mixing assumption is reduced by the fact that snowpack is typically homogenised during snowmelt when diurnal melt/refreeze processes take place in the snowpack (Taylor et al. 2001, Unnikrishna et al. 2002, Koeniger et al. 2008). The majority of snowpack outflow is generated during the overall snowmelt when isotopes in the snowpack are subjected to mixing, which gives empirical ground to our simplification. The limitations of the snow isotope modelling regarding the full-mixing assumption and potential biases caused by rain-on-snow events and blowing wind redistribution are further discussed in parallel work by Ala-aho et al (in review). In that study we also provide further evidence for the usefulness snow isotope modelling approach by finding a good agreement between simulated snowmelt isotopes and snowmelt lysimeter data sampled in Bogus and Krycklan. With the present study we demonstrate that even with the relatively simple isotope model we are able to produce improved estimates of spatially distributed snowmelt isotopes, which is called for in tracer-aided modelling of sparsely monitored snow-influenced regions (Smith et al. 2016, Delavau et al. 2017). Furthermore, we show that the stream isotopes can be used to inform parameter the snow routine through calibration, in particular for the snow sublimation fractionation."*

P7, L22. It appears that the model uses a constant value for evaporation fractionation factor, whereas it is expected to vary with meteorological conditions such as relative humidity and wind speed. Please present justification for using a constant value. P7, L20. Snow loss to the atmosphere occurs by two different processes depending on snow surface temperature; sublimation under 0 C, and evaporation of melt water at 0 C. Resulting isotopic fractionation factors may be different. This is a subtle point, but should be discussed.

Again, we thank the reviewer for this good observation of the model simplification. Both of the comments above, regarding the constant value for fractionation factor and differentiation between evaporation/sublimation fractionation, will be addressed by discussing approach we chose by adding the following paragraph:

*"…In typical winter conditions with low air temperature and high relative humidity, we would expect the equilibrium fractionation to dominate over kinetic fractionation (Gat and Gonfiantini 1981), therefore making weather conditions and the differentiation between the two processes of lesser importance. We also did not differentiate between sublimation (ice to vapour) and evaporation (liquid water retained in the snow to vapour). Liquid water evaporation has a smaller equilibrium fractionation factor (3.5 ‰) compared to sublimation (15 ‰), so separating the different processes could lead to smaller simulated fractionation signal. In our approach we lumped the above*

*fractionation processes and their temporal variability caused by meteorological conditions in the $E_{frac}$ calibration parameter with the purpose of keeping the simulated isotope process complexity to a minimum, which is in line with our conceptual modelling of water in the catchment. The simplified approach is further justified by the limited power of the validation data (isotopes in streamflow) to constrain the additional parameters required for more sophisticated snowpack isotope modelling methods (Taylor et al. 2001)."*

P7, L26. What is the reasoning for dividing M_frac by d_melt? Is this purely empirical or does it have a theoretical basis?

A good question: the basis of the formulation is in accumulated field evidence of gradual snowmelt and snowpack enrichment during snowmelt. However, the equation is in essence empirical. We suggest to clarify/highlight this with the following sentence:

*"The empirical formulation in Eq. (3) is proposed in order to mimic the gradual isotopic enrichment of both snowmelt runoff and snowpack over the overall melt period, which is frequently observed in field studies (Taylor et al. 2002) and theoretically show in modelling experiments (Feng et al. 2002)"*

P8, L1. I am generally against citing unaccepted "in review" manuscript because there is no guarantee that the manuscript will be published. Please avoid using the in review manuscript, or at least include it in the reference list with the journal name.

We agree and we understand this critique, and we appreciate the inconvenience and difficulty of having the two interlinked papers in review simultaneously. Unfortunately, in the current work, it was clear that the two papers were impossible to integrate, and the work was carried out simultaneously. However, the cited paper is currently accepted, subject to revisions, in Water Resources Research. We hope the paper is accepted in the coming days, however, if it is not fully accepted before the final revision of this work, the citation given here will be:

*"Ala-aho, P., Tetzlaff, D., McNamara, J. P., Laudon, H., Kormos, P. and Soulsby, C.: Modelling the isotopic evolution of snowpack and snowmelt: testing a spatially distributed parsimonious approach, accepted in Water Resour. Res., subject to revisions, in review."*

P8, L3-4. What kind of algorithm is used for the new snow module? Is it still a degreeday model? I note that radiation is included in the data set (P5, L32)? Please include a brief explanation of the model. Note that in a catchment with rugged topography such as Bogus, slope aspect and angle may have a strong influence on the spatial distribution of snow accumulation and melt.

We agree that more detailed description of the snow module would be helpful, but we would preferably leave the full equations to be presented in Ala-aho et al. (in review) to keep the paper length reasonable and to avoid overlap. To better explain the concepts of the snow model, we suggest to add:

*"Energy balance for each time step is solved based on net radiation, latent and sensible heat, heat advection from precipitation and heat storage in the snowpack. The energy balance is coupled with mass balance equations solving the amount and ice, and liquid water retained in the snowpack and the snowmelt and sublimation fluxes. Model inputs for precipitation and air temperature are spatially distributed as described in section 2.2, and the radiation terms are adjusted to the influence of slope, aspect, hillshading, and canopy sheltering. Tree canopy snow interception and unloading are simulated after (Hedstrom and Pomeroy 1998)."*

P8, L6. This is another questionable assumption. Snow accumulates from the bottom to the top during winter without much mixing. Snow melt and evaporation occurs from the top. Therefore, it is questionable to assume complete mixing for water age. Please point this out in texts and discuss potential errors resulting from this assumption.

We respectfully argue that full mixing of the snow age is not a poor assumption in the context of the aims of this model, as it is the age of snowmelt water entering the catchment that we are really trying to constrain. We will clarify this by adding:

"*With the full-mixing assumption, water stored as snow is aged while the snowpack persists, but this is refreshed with new snowfall, weighted by snow amount. Water entering the catchment during snowmelt will therefore be reasonably approximated as having an age younger than the full snow-covered season, but considerably older than only the most recent snowfall. Therefore the snowmelt entering the catchment is typically older than precipitation, depending on the length of season of snow-coverage*"

P8, L8-10. I do not quite understand this sentence. How is FC defined? By calibration? It is not listed in Table 1.

We will rephrase the sentence as follows, which defines the field capacity more clearly:

Original: "The concept of field capacity also is changed from where the field capacity (FC) was the maximum amount of water that could be stored in the linear soil storage (SM) to a fraction of the total storage volume ($SM_{max}$)."

Revised: "*The concept of field capacity also is changed from Huijgevoort et al (2017a) from where the field capacity (FC) was defined as the maximum amount of water that could be stored in the linear soil storage (SM). Now we conceptualise this - as more typically done in soil physics - as the amount of water that is preferably retained in the soil, defined by parameters for volumetric field capacity and soil depth (Eq. A9), both technically measurable in the field.*"

P8, L11. The need for adaptation became apparent. How?

A sentence will be added to elucidate the field conditions to which the model parameterisation needed to be adjusted:

"*In its original formulation the model did not allow for high enough seepage rates from the soil to groundwater domain as observed in Bogus, or non-linearly increased runoff generation from the soil domain during times of high soil storage, also known as the transmissivity feedback, present in Krycklan.*"

P8, L13. Depending on the soil thickness, root density, and other complex factors, it is unlikely that evaporation age is equal to average soil water age. Please discuss this issue carefully.

A fair comment. Justification for the simplifying the evaporated water age simulation from previous model iteration, and suggestions for better representing evaporation age will be added as follows:

"*Finally, in contrast to previous work, here we assumed the evaporation age to be equal to the water age in the soil storage. In the previous model implementation in Huijgevoort et al. (2016a), the simulated soil water age was affected by evaporation, but the simulated isotope composition of the soil was not; as a result the simulated evaporation age was not informed/constrained by the isotope model calibration and in this study it was excluded to simplify the model. Re-incorporating evaporated water age in the simulations would benefit from vertically layered soil parameterisation*"

*and from explicit hydrological partitioning between evaporation and transpiration (see e.g. Sprenger et al. 2016)."*

P8, L16. There appear to be 13 parameters listed in Table 1. If three values are used for each of the 13 parameters, there are roughly 1.6 million combinations (3ˆ13) of parameter values. How were the 10,000 combinations selected?

Following information about the model's Monte Carlo calibration will be added:

*"We used random sampling of the parameter space assigning uniform distribution for all parameters, with pre-defined parameter ranges (minimum and maximum parameter value) given in Table 1. The model was run 10 000 times, each run with a different, randomly sampled parameter set."*

Please explain. P8, L20. Were empirical coefficients used in the snow module (e.g. coefficient for degree-day model)? If so, how were they calibrated? Please explain.

The only four model parameters calibrated for the snow module were detailed in the text (P8L20), and the calibration of all model parameters is explained in the same section. The snow module is not a degree-day model, but a process-based single layer energy and mass balance model, which is now better explained in relation to an earlier comment.

P8, L25. Is this field capacity the same FC as the one described in L8-10 above?

Yes, abbreviation (FC) will be added here to clarify

P9, L17. The standard metric for stream flow calibration is Nash-Sutcliff efficiency (NSE), which is familiar to most of the readers of this journal. I do not see a strong justification for using Kling-Gupta efficiency (KGE). I suggest NSE be used instead of KGE. If not, please present a stronger justification for using KGE and include its definition (e.g. equation).

Thank you for the suggestion. Equations for the goodness-of-fit metrics will be provided:

$$KGE = 1 - \sqrt{(r-1)^2 + (\mu_s/\mu_o - 1)^2 + (\sigma_s/\sigma_o - 1)^2} \qquad (4)$$

$$MAE = \frac{\sum_{i=1}^{N} |s_i - o_i|}{N} \qquad (5)$$

Where r: Pearson correlation coefficient; μ: the mean; σ: the standard deviation; subsripts s and o refer to simulated and observed values, respectively, N: number of simulation-observation pairs

We respectfully propose to continue using KGE, and will attempt to better justify its selection as follows:

*"Kling-Gupta efficiency was used because it combines several measures of misfit between observations and simulations (correlation, bias and a measure of relative variability; first, second and third term inside the square root in Eq. 4, respectively) into a single number in a more robust way than the frequently used Nash-Sutcliffe performance metric (Gupta et al. 2009). The selected GOF metric ultimately remains a subjective choice in any model calibration, but with the considerations above we found the selected metrics facilitating convenient and robust comparison between catchments. Additionally, the NSE puts a primacy on simulation of high flows, whereas, for a hydrological model to accurately and simultaneously capture isotope dynamics across the flow regime, a more balanced GOF measure for stream flows is needed as shown in other studies (e.g. Birkel et al., 2015)"*

P9, L28 – P10, L3. I read this section several times, and still could not understand the definition of F ( ) and how it was constructed. Please re-write the section more clearly.

We acknowledge that the model calibration method is unconventional and therefore needs further clarification. To do so we propose a paragraph explaining the calibration procedure less mathematically:

*"To clarify the calibration procedure with an example, let's consider two GOF measures, KGE of streamflow and SWE, to constrain the selection of 100 behavioural simulations from an ensemble. In this case it is unlikely, although possible, that the same 100 simulations that produce the highest GOF values for streamflow would also have the highest GOF values for SWE. To find the threshold quantile above which the GOF from exactly 100 runs in both calibration objectives map, first an initial guess is made; we used $F_{Xf}(x_f) = F_{Xs}(x_s) = 0.5$, which corresponds to the median of GOF values for $x_f$ and $x_s$, streamflow and SWE, respectively. This quantile as a threshold it is checked how many individual simulations produce GOF values that are higher than $x_f$ and $x_s$ for both streamflow and SWE, respectively. If the number of simulations above the $x_f$ and $x_s$ GOF thresholds in both objectives is higher than the preassigned number $n_{run}$ (in our case 100), a step up the CDF is taken, by adding a small increment in the threshold quantile, for example: $F_{Xf}(x_f) = F_{Xs}(x_s) = 0.51$. Then the number of simulations for which KGE value is exceeded $x_f$ and $x_s$ for both streamflow and SWE are again counted for the updated threshold, and the process is repeated, until a quantile $F_{Xf}(x_f) = F_{Xs}(x_s)$, for which $n_{run}=100$ is reached. The resulting threshold GOF value $x_f$, in this example measured in KGE, will be lower than if constrained by flow data alone, because some simulations producing a good KGE for flows will be rejected as they have a $GOF_f < x_s$ for SWE."*

P10, L6-9. This section was also very difficult to follow. Please re-write.

The section will be reformulated as follows, and more information is added to discuss the motivation of the calibration technique:

*"The introduced approach allows pre-specifying the number of behavioural runs while circumventing the need to combine the GOF metrics into a single objective function (e.g. (Huijgevoort et al. 2016). When a single objective function is constructed from multiple GOF metrics, it is often difficult to combine GOF metrics that need to be maximised (such as KGE) and minimised (such as MAE) in the model calibration. Our approach is based on quantiles of the GOF metric rather than its numerical value, making the method convenient in combining metrics that are to be minimised or maximised, or have different ranges of numerical values."*

P10, L30-31. The model is calibrated for stream isotopic composition, but it is not clear how well the isotopic compositions in groundwater and soil water are simulated. Laudon et al. (2013) describes systematic sampling programs for soil water and groundwater at Krycklan catchment. Please include a comparison of simulated and observed soil water and groundwater data, where available.

Thank you for the insightful suggestion, which would be a useful the diagnostic of model performance. Such comparison with STARR simulated and observed soil and groundwater data is done in the first STARR model application in Huijgevoort et al. (2016a). In this work similar analysis could be done, however the data would be inconsistent among the study catchments. Furthermore we feel that the analysis it would involve would excessively lengthen the results section without contributing substantially to the main outcomes of the study. We suggest to add the following to acknowledge the possibility for using additional data for model calibration:

*"The spatially distributed model structure would allow further model testing using internal model variables, such as soil and groundwater and snowmelt isotope composition, as done successfully for*

*the sites in Huijgevoort at al. (2016a) and Ala-aho et al. (in review), respectively. However, with the focus on catchment comparison in this study we restrict our analysis to the stream isotopes, which have been sampled in all study sites."*

P15, L2. It is true that STARR has a spatially distributed model structure, but it uses uniform values of model parameters for an entire catchment. In reality, the spatial distribution of material properties (e.g. soil type and thickness, bedrock type, vegetation) has an important effect on catchment hydrological responses. That is why many models use hydrological response unit (HRU) approaches to capture catchment heterogeneity. I suggest that the authors discuss the lack of heterogeneity in the current configuration of STARR and its potential implication for model performance and estimated water age.

We appear to have inadequately explained that where there was sufficient data to spatially vary parameters (for soil and vegetation), we did. We will add the following section to the discussion:

*"We used spatially varied parameterisation for soil properties and vegetation where there was sufficient data to do so; a differentiation between mineral and organic soil was made in Bruntland and Krycklan, a detailed soil depth map was used in Bogus and vegetation LAI was estimated form either vegetation maps (Krycklan) or three height (Bruntland and Bogus). Even so, naturally occurring small-scale heterogeneity is known to influence the catchment hydrological response (Beven and Germann 1982), but it is difficult to represent in hydrological models - one of the persistant problems in hydrological modelling (Blöschl and Sivapalan 1995, Beven 2002). Every new introduced element of heterogeneity typically comes with a burden of increased number of parameters (see soil parameterisation in Table 1) which can lead to model equifinality issues (Beven 2006). We opted to minimise the number of calibrated parameters, with the trade-off off bringing spatial variability in parameter values only when supported by field data."*

P16, L10. High sensitivity of E_frac in Fig. 10. This is a bit misleading because the highest sensitivity is observed for Bogus catchment, which had few data points. Fig. 10 indicates low sensitivity for E_frac for Bruntland catchment. If the authors want to showcase the snow isotope model as one of the highlights of this work, then this topic needs to be explored a bit more carefully. Please note my comments on the isotopic fractionation in P7, L10.

Thank you for the perceptive comment. We readily discussed the reasons for the $E_{frac}$ parameter sensitivity in Krycklan and Bogus P16L9-21.To add to that, we will add reasoning for the low sensitivity in Bruntland:

*"The $E_{frac}$ parameter was insensitive for the Bruntland (Fig. 10), which is not surprising given the considerably smaller snow-influence compared to the other two sites (Fig. 6)."*

In addition, we will soften the wording of the importance of snow sublimation processes by modifying the abstract:

Original: "Our study demonstrated the importance of including snow evaporative fractionation processes in tracer-aided modelling for catchments with seasonal snowpack…"

Revised: "*Our study suggested that snow sublimation fractionation processes can be important to include in tracer-aided modelling for catchments with seasonal snowpack…*"

P17, L8. I do not think isotope data from Bogus catchment had exceptionally high quality. Please clarify.

We see the point, "exceptionally high quality" will be removed.

P17, L11. I do not think the need to incorporate isotopic evaporative fractionation is convincingly demonstrated. Please see my comment on Fig. 10 above.

Following the reviewer suggestion we will tone down the focus away from the snow isotope module, and remove the following sentence from the conclusions:

*In particular, we were able to demonstrate the need to incorporate isotope evaporative fractionation processes in seasonal snowpack*

Fig. 5. It is impossible to see the difference between red line and pink band in the top figure. Can you use different color combination (e.g. blue and pink) and use the same color for all of Fig. 5, 6, and 7? I do not see a real need for using different colors for different catchments.

We appreciate the difficulty of differentiating the colours showing the median and the range of behavioural simulations. However, testing showed that the problem is not greatly alleviated by different colour scheme, because the min-max range is narrow at times. The purpose of the colour red/green/blue colour coding in figs 2,3,5,6,7, 8 and 10 is to have the same colour representing a given site, which we hope will guide the reader. This will be pointed out in the caption of Fig. 2, at the first instance where the colour coding is used:

*"Colour coding: red for Krycklan, blue for Bruntland, green for Bogus are maintained through the manuscript."*

Fig. 9. Please include a scale and a north arrow for each catchment.

Scale and north arrow are present in Fig.1, and they are shared between the catchments, which was readily pointed out in the caption for Fig.1.

List of additional references proposed:

Beven, K.: Towards an alternative blueprint for a physically based digitally simulated hydrologic response modelling system, Hydrol. Processes, 16, 189-206, 2002.

Beven, K. and Germann, P.: Macropores and water flow in soils, Water Resour. Res., 18, 1311-1325, 1982.

Beven, K.: A manifesto for the equifinality thesis, J. Hydrol., 320, 18-36, 2006.

Blöschl, G. and Sivapalan, M.: Scale issues in hydrological modelling: a review, Hydrol. Process., 9, 251-290, 1995.

Dahlke, H. E. and Lyon, S. W.: Early melt season snowpack isotopic evolution in the Tarfala valley, northern Sweden, Ann. Glaciol., 54, 149-156, 2013.

Feng, X., Taylor, S., Renshaw, C. E. and Kirchner, J. W.: Isotopic evolution of snowmelt 1. A physically based one-dimensional model, Water Resour. Res., 38, 35-1-35-8, 2002.

Gat, J. R. and Gonfiantini, R.: Stable isotope hydrology. Deuterium and oxygen-18 in the water cycle, IAEA, Vienna, 1981.

Goody, R. M. and Yung, Y. L.: Atmospheric radiation: theoretical basis, Oxford University Press, 1995.

Rodhe, A.: Spring flood meltwater or groundwater?, Hydrology Research, 12, 21-30, 1981.

Sevruk, B. and Mieglitz, K.: The effect of topography, season and weather situation on daily precipitation gradients in 60 Swiss valleys, Water science and technology, 45, 41-48, 2002.

Sprenger, M., Seeger, S., Blume, T. and Weiler, M.: Travel times in the vadose zone: Variability in space and time, Water Resour. Res., 52, 5727-5754, 2016.

Stone, P. H. and Carlson, J. H.: Atmospheric lapse rate regimes and their parameterization, J. Atmos. Sci., 36, 415-423, 1979.

Unnikrishna, P. V., McDonnell, J. J. and Kendall, C.: Isotope variations in a Sierra Nevada snowpack and their relation to meltwater, J. Hydrol., 260, 38-57, 2002.

---

## Author Comment (AC3) · 1 Jun 2017

**Author response to Referee #3 comments**

Manuscript hess-2017-106, "Using isotopes to constrain water flux and age estimates in snow-influenced catchments using the STARR (Spatially distributed Tracer-Aided Rainfall-Runoff) model" by Ala-aho, P. et al.

We are grateful for the remarks by Referee #3 on our manuscript. We have addressed the comments providing suggestions for corrections and clarification as requested. We hereby provide our point by point responses how the comments by ref #3 will be addressed in the revised manuscript.

Sincerely,

Pertti Ala-aho

**Referee #3 comments**

The article developed a calculation scheme to describe the spatially and temporally variable isotope fractionation processes in seasonal snowpacks, and the calculation scheme was linked to the STARR model to simulate the stream flows, isotope ratios and snow pack dynamics in three long term experimental catchments.

GENERAL COMMENTS:

1. The spatially distributed water age was simulated as shown in Fig.9. Does the spatially distributed water age is estimated by averaging the water age of surface water, soil stored water and groundwater stored water in each computing unit? The water age of incoming precipitation is taken as 1, which means the water age of stored water in the catchment should be larger than 1. Why the calculated water age is less than 1, even equal to 0, in many places of the three catchments?

Thank you for the comment, we realise that the legend in Fig. 9 can cause confusion. The age of 0, pointed out by the referee, indicated by the dark blue colour, means water with age between 0-0.5 years. The legend in Fig. 9 will be changed to make the "age bins" more obvious.

SPECIFIC COMMENTS:

1. L18P1, P4 It is better to specify how long does the "exceptionally long" mean?

To give better context to your claim for "exceptionally long datasets" we will rephrase this to highlight that the datasets are exceptional in the context of Northern catchments, rather than specifying data record lengths (which vary for each catchment and observation):

*"In the context of Northern catchments the sites have exceptionally long and rich datasets of hydrometric data and - most importantly - stable water isotopes for both rain and snow conditions."*

The observation period for the hydrometric data and the stable water isotopes date should also be given in the "Study sites" section for better understanding the simulation process and calibration results.

Thank you for the suggestion, however we respectfully suggest to leave the data description in its own section. We find that merging the site and data description would lead to a more fragment structure, and make the comparison between sites in section "2.2 Model input and test data" more difficult.

2.L24P1, The geology setting is only briefly described for the Bogus Creek, but not given for the other two study sites in the article. How to make a conclusion that the geology is an important factor causing contrasting water age distribution among the three catchments?

A fair comment, we will remove the term geology and refer only to soil characteristics

3. L8P2, The meaning of abbreviation of "STARR" has been given in L2P2.

Thank you for pointing this out, repetition will be removed.

4. L12P2, The literature "Ala-aho et al., in review" has been cited several times in the article. It's better to give a specific published journal or publishing house etc. of the literature.

It is very likely that the paper will be soon in press (it is accepted subject to revision) if it is not accepted before the HESS paper is published the citation will be changed to:

*Ala-aho, P., Tetzlaff, D., McNamara, J. P., Laudon, H., Kormos, P. and Soulsby, C.: Modelling the isotopic evolution of snowpack and snowmelt: testing a spatially distributed parsimonious approach, accepted in Water Resour. Res., subject to revisions, in review.*

5. L25P2, The sentence seems written in Latin.

This text will be removed

6. L26P5, The meaning of DCEW should be given here.

DCEW will be defined at its first occurrence

7. L33P5-L5P6, The meteorological date from stations in the neighboring catchments are used for the Krycklan and Bruntland Burn catchments. It's better to describe the distance of the stations from the study sites, or give the latitude and longitude coordinates of the stations for better judging the reliability of the inputted meteorological data in the model.

Distances between meteorological stations and the catchment outlets will be added to the text

8. L31P8, What does the "kg" means? Hydraulic conductivity?

The sentence will be rephrased to make the meaning of $k_g$ clear:

*"For the groundwater one parameter ($k_g$), which linearly relates the groundwater storage to groundwater outflow ($Q_{gw}$), was calibrated."*